

**The impact of aged wildfire smoke on atmospheric composition and**
**ozone in the Colorado Front Range in summer 2015**
Jakob Lindaas[1], Delphine K. Farmer[2], Ilana B. Pollack[1,2], Andrew Abeleira[2], Frank Flocke[3], Rob
Roscioli[4], Scott Herndon[4], and Emily V. Fischer[1]
[1] Colorado State University, Department of Atmospheric Science, Fort Collins, CO, USA
[2] Colorado State University, Department of Chemistry, Fort Collins, CO, USA
[3] National Center for Atmospheric Research, Boulder, CO, USA
[4] Aerodyne Research Inc., Billerica, MA, USA
*Correspondence to*: Jakob Lindaas (jlindaas@rams.colostate.edu) or Emily V. Fischer (evf@rams.colostate.edu)
**Abstract.** The relative importance of wildfire smoke for air quality over the western U.S. is expected to increase as the
climate warms and anthropogenic emissions decline. We report on *in situ* measurements of ozone ($O_3$), a suite of volatile
organic compounds (VOCs), and reactive oxidized nitrogen species collected during summer 2015 at the Boulder
Atmospheric Observatory (BAO) in Erie, CO. Aged wildfire smoke impacted BAO during two distinct time periods during
summer 2015: 6 – 10 July and 16 – 30 August. The smoke was transported from the Pacific Northwest and Canada across
much of the continental U.S. Carbon monoxide and particulate matter increased during the smoke-impacted periods, along
with peroxyacyl nitrates and several VOCs that have atmospheric lifetimes longer than the transport timescale of the smoke.
During the August smoke-impacted period, nitrogen dioxide was also elevated during the morning and evening compared to
the smoke-free periods. There were six days during our study period where the maximum 8-hour average $O_3$ at BAO was
greater than 65 ppbv, and two of these days were smoke-impacted. We examined the relationship between $O_3$ and
temperature at BAO and found that for a given temperature, $O_3$ mixing ratios were greater (~10 ppbv) during the smoke-
impacted periods. Enhancements in $O_3$ during the August smoke-impacted period were also observed at two long-term
monitoring sites in Colorado: Rocky Mountain National Park and the Arapahoe National Wildlife Refuge near Walden, CO.
Our data provide a new case study of how aged wildfire smoke can influence atmospheric composition at an urban site, and
how smoke can contribute to increased $O_3$ abundances across an urban-rural gradient.

**Keywords.** wildfire smoke, air quality, ozone, *in situ* observations, biomass burning
**1 Introduction**
Over the past 30 years, wildfires in the western U.S. have increased in both frequency and intensity, and this trend will likely
continue under future climate change (Westerling, 2016). Wildfire smoke can be transported over thousands of kilometers,





and exposure to wildfire smoke has significant impacts on human health (Künzli et al., 2006; Rappold et al., 2011; Elliott et
al., 2013). While U.S. emissions of most major air pollutants are declining (Pinder et al., 2008), increasing fire activity
suggests that wildfires may have a greater relative impact on U.S. air quality in the future (Val Martin et al., 2015).

Ozone ($O_3$) is formed when hydrocarbons are oxidized in the presence of nitrogen oxides ($NO_x = NO + NO_2$) and sunlight
(Sillman, 1999). Wildfires emit many trace gas species that contribute to tropospheric $O_3$ production. Along with carbon
monoxide (CO), methane ($CH_4$), and carbon dioxide ($CO_2$), hundreds of different non-methane volatile organic compounds
(NMVOCs) with lifetimes ranging from minutes to months (Atkinson and Arey, 2003) are emitted during biomass burning
(Akagi et al., 2011; Gilman et al., 2015). Due to relatively large emissions of $CO_2$, CO, $CH_4$ and $NO_x$, the contribution of
VOCs to the total emissions from fires on a molar basis is small (<1%). However, VOCs dominate the OH reactivity in
smoke plumes (Gilman et al., 2015). Recent observations of the evolution of VOCs within aging smoke plumes indicate that
OH can be elevated in young biomass burning plumes (Hobbs et al., 2003; Yokelson et al., 2009; Akagi et al., 2012; Liu et
al., 2016) in part due to the photolysis of oxygenated VOCs (Mason et al., 2001), which make a large contribution to the
total emitted VOC mass (Stockwell et al., 2015). Elevated OH may reduce the lifetime of emitted VOCs and increase
oxidation rates and potential $O_3$ production.

Fires are also a major source of oxidized nitrogen; emissions from biomass and biofuel burning represent approximately
15% of total global $NO_x$ emissions (Jaegle et al., 2005).  However, there are major uncertainties in $NO_x$ emission estimates
from biomass burning, particularly at a regional scale (Schreier et al., 2015). $NO_x$ emissions depend on the nitrogen content
of the fuel (Lacaux et al., 1996; Giordano et al., 2016) as well as the combustion efficiency (Goode et al., 2000; McMeeking
et al., 2009; Yokelson et al., 2009). Emitted $NO_x$ is quickly lost in the plume, either by conversion to $HNO_3$ (Mason et al.,
2001) or via PAN formation (Alvarado et al., 2010; Yates et al., 2016). $HNO_3$ is not often observed in plumes because it
either rapidly forms ammonium nitrate or is efficiently scavenged by other aerosols (Tabazadeh et al., 1998; Trentmann et
al., 2005).

There are multiple lines of observational evidence indicating that wildfires in the western U.S. increase the abundance of
ground level $O_3$. Background $O_3$ mixing ratios across the western U.S. are positively correlated with wildfire burned area
(Jaffe et al., 2008), and daily episodic enhancements in $O_3$ at ground sites can be > 10 ppbv (Lu et al., 2016). There are well-
documented case studies of within plume $O_3$ production (Jaffe and Wigder, 2012) and time periods where smoke contributed
to exceedances of the U.S. EPA National Ambient Air Quality Standard (NAAQS) for $O_3$ (Morris et al., 2006; Pfister et al.,
2008). Brey and Fischer (2016) investigated the impacts of smoke on $O_3$ abundances across the U.S. via an analysis of
routine *in situ* measurements and NOAA satellite products. They found that the presence of smoke is correlated with higher
$O_3$ mixing ratios in many areas of the U.S., and that this correlation is not driven by temperature.  Regions with the largest





smoke-induced $O_3$ enhancements (*e.g.* the southeast and Gulf coast) can be located substantially downwind of the wildfires
producing the most smoke.

Despite several recent studies showing that smoke contributes to elevated $O_3$, there have been relatively few detailed studies
of wildfire smoke mixing with anthropogenic air masses near the surface. Singh et al. (2012) used aircraft measurements
from summer 2008 over California to document significant $O_3$ enhancements in nitrogen-rich urban air masses mixed with
smoke plumes. Accompanying air quality simulations were not successful in capturing the mechanisms responsible for these
enhancements. In general, measurements of $O_3$ precursors are hard to make routinely. Instrumentation and calibration
methods tend to be time and labor intensive, and thus unpredictable wildfire smoke plumes and their effects on surface $O_3$
are sparsely sampled.

Here we present a case study of aged wildfire smoke mixed with anthropogenic pollution in the Colorado Front Range and
show its impact on atmospheric composition and $O_3$. This region violates the NAAQS for $O_3$, and has been the focus of
several recent studies (*e.g.* McDuffie et al., 2016; Abeleira et al., 2017). First we describe the research location and
measurements. Next, we identify the smoke-impacted time periods and show the origin, approximate age, and wide
horizontal extent of the smoke plumes. We characterize significant changes in atmospheric composition with respect to the
two major classes of $O_3$ precursors, VOCs and oxidized reactive nitrogen ($NO_y$). Finally, we present the impact of smoke on
$O_3$ abundances during this period and discuss the underlying causes of this impact.
**2 Measurements and Research Site**
During summer 2015, we made measurements of a suite of trace gases at the Boulder Atmospheric Observatory (BAO),
located north of Denver, CO, in the middle of the rapidly developing northern Colorado Front Range [40.05˚N, 105.01˚W,
1584m ASL]. BAO has a history of atmospheric trace gas and meteorological measurements stretching back nearly four
decades (Kelly et al., 1979; Gilman et al., 2013). Our research campaign from 1 July – 7 September 2015 measured a suite
of $O_3$ precursor species as well as several $NO_x$ oxidation products and greenhouse gases. The intended goal of the field
campaign was to improve our understanding of the complex $O_3$ photochemistry in the Colorado Front Range and the
contributions of oil and natural gas activities as well as other anthropogenic emissions to $O_3$ production. All measurements
were made by instruments housed in two trailers located at the base of the BAO tower. Here we briefly describe the
measurements used in this paper. Data are available at https://esrl.noaa.gov/csd/groups/csd7/measurements/2015songnex/.

We measured CO and $CH_4$ at ~3 second time resolution with a commercial cavity ring-down spectrometer (Picarro, model
G2401) (Crosson, 2008). The inlet was located 6 m above ground level (a.g.l.), and a 1 μm PTFE filter membrane (Savillex)
at the inlet was changed weekly. Laboratory instrument calibrations were performed pre- and post-campaign using three





NOAA standard reference gases (http://www.esrl.noaa.gov/gmd/ccl/refgas.htmls; CA06969, CB10166, and CA08244). Field
calibration was performed every 3 hours using high, low and middle reference gas mixtures (Scott Marin Cylinder IDs
CB10808, CB10897, CB10881). Mixing ratios were calculated using the WMO-CH4-X2004 and WMO-CO-X2014 scales.
The uncertainty associated with the $CH_4$ and CO data is estimated to be 6% and 12% respectively, and it was estimated as
the quadrature sum of measurement precision, calibration uncertainty and uncertainty in the water vapor correction.

A custom 4-channel cryogen free gas chromatography (GC) system (Sive et al., 2005) was used to measure selected non-
methane hydrocarbons (NMHCs), $C_1 - C_2$ halocarbons, alkyl nitrates (ANs), and oxygenated volatile organic compounds
(OVOCs) at sub-hourly time resolution. The inlet was located at 6 m a.g.l. with a 1 µm teflon filter. A calibrated whole air
mixture was sampled in the field after every ten ambient samples to monitor sensitivity changes and measurement precision.
A full description of this instrument is provided in (Abeleira et al., 2017).

Ozone data at BAO for this time period were provided by the NOAA Global Monitoring Division surface ozone network
(McClure-Begley et al., 2014; data available at aftp.cmdl.noaa.gov/data/ozwv/SurfaceOzone/BAO/). Ozone was measured
via UV-absorption using a commercial analyzer (Thermo-Scientific Inc., model 49), which is calibrated to the NIST standard
over the range 0 – 200 ppbv and routinely challenged at the site. The inlet height was 6m a.g.l. on the BAO tower, located
about 50 feet from the two trailers, and measurements were reported at a 1 minute averaging interval with an estimated error
of 1%.

Nitrogen oxides ($NO_x \equiv NO+NO_2$) and total reactive nitrogen ($NO_y$) were measured via $NO-O_3$ chemiluminescence detection
(Kley and McFarland, 1980) using a commercial analyzer (Teledyne, model 200EU). Two commercial converters, a 395 nm
-LED converter (Air Quality Designs, Inc., model BLC) for chemically-selective photolysis of $NO_2$ to NO and a
molybdenum in stainless steel converter (Thermo Scientific Inc.) heated to 320 ºC for reduction of $NO_y$ to NO, were
positioned as close to the inlet tip as possible (<10 cm). A 7 µm stainless steel particulate filter was affixed to the upstream
end of the molybdenum converter; otherwise no other filters were used. The analyzer switched between sampling from the
LED ($NO_x$) converter and the molybdenum (NOy) converter every 10 seconds, and the LEDs were turned on (to measure
$NO+NO_2$) and off (to measure NO only) every minute. $NO_2$ was determined by subtraction of measured NO from measured
$NO+NO_2$ divided by the efficiency of the LED converter. All three species are reported on a consistent two-minute average
timescale. The detector was calibrated daily by standard addition of a known concentration of NO, NIST-traceable (Scott-
Marrin Cylinder ID CB098J6), to synthetic ultrapure air. Both converters were calibrated with a known concentration of $NO_2$
generated via gas phase titration of the NO standard. The $NO_y$ channel was further challenged with a known mixing ratio of
nitric acid ($HNO_3$) generated using a permeation tube  (Kintech, 30.5 ± 0.8 ng/min at 40 ºC), which was used to confirm
>90% conversion efficiency of $HNO_3$ by the molybdenum converter. Uncertainties of ±5% for NO, ±7% for $NO_2$, and ±20%



for $NO_y$ are determined from a quadrature sum of the individual uncertainties associated with the detector, converters, and
calibration mixtures; an LOD of 0.4 ppbv for all species is dictated by the specifications of the commercial detector.

Peroxyacyl nitrates (PANs) were measured using the National Center for Atmospheric Research gas chromatograph with an
electron capture detector (NCAR GC-ECD) (Flocke et al., 2005). The instrument configuration was the same as was used
during the summer 2014 FRAPPE field campaign (Zaragoza, 2016). The NCAR GC-ECD analyzed a sample every five
minutes from a 6 m a.g.l. inlet with 1μm teflon filter. A continuous-flow acetone photolysis cell generated a known quantity
of PAN used to calibrate the system at 4-hour intervals.

An Aerodyne dual quantum cascade laser spectrometer was used to measure $HNO_3$ (McManus et al., 2011). The instrument
employed a prototype 400 m absorption cell for increased sensitivity during the first month of the campaign, after which it
was replaced by a 157 m absorption cell. An active passivation inlet (Roscioli et al., 2016) was used to improve the time
response of the measurement to ~0.75 s. This technique utilized a continuous injection of 10-100 ppb of a passivating agent
vapor, nonafluorobutane sulfonic acid, into the inlet tip. The inlet tip was made of extruded perfluoroalkoxy Teflon (PFA),
followed by a heated, fused silica inertial separator to remove particles larger than 300 nm from the sample stream. The inlet
was located 8 m a.g.l. with a 18 m heated sampling line (PFA, 1/2" diameter OD) to the instrument. The system was
calibrated every hour by using a permeation tube that was quantified immediately prior to the measurement period.
**3 Smoke Events**
We observed two distinct smoke-impacted periods at BAO, identified by large enhancements in CO and fine aerosol ($PM_{2.5}$).
Figure 1 presents CO observations from BAO and fine particulate matter ($PM_{2.5}$) observations from the Colorado
Department of Public Health and Environment (CDPHE) CAMP air quality monitoring site, located in downtown Denver,
approximately 35km south of BAO. $PM_{2.5}$ was similarly elevated during the smoke-impacted periods at CDPHE monitoring
sites across the Colorado Front Range (not shown). For our analysis, we defined a July smoke-impacted period and an
August smoke-impacted period. The July smoke-impacted period lasted for 4 days from 00 MDT 6 July 2015 to 00 MDT 10
July 2015. The August smoke-impacted period was significantly longer (~14 days). For the subsequent analysis, we
combined three distinct waves of smoke-impact in this 14 day period into one August smoke-impacted period: 00 MDT 16
August 2015 – 18 MDT 21 August 2015, 12 MDT 22 August 2015 – 18 MDT 27 August 2015, and 14 MDT 28 August
2015 – 09 MDT 30 August 2015. We omitted the brief periods between these times from the analysis due to uncertainty on
the influence of smoke during them. All other valid measurements were considered part of the smoke-free data.

Figure 2 presents the extent of the presence of smoke in the atmospheric column during representative smoke-impacted days,
7 July and 21 August 2015. The NOAA Hazard Mapping System smoke polygons (grey shading) show that the smoke



events observed at BAO were large regional events.  The HMS smoke product is produced using primarily visible satellite
imagery (Rolph et al., 2009). The extent of smoke plumes within the HMS dataset represents a conservative estimate, and no
information is provided on the vertical extent or vertical placement of the plumes. The red triangles represent MODIS active
fire locations for the previous day (Giglio et al., 2003; Giglio et al., 2006). The thin black lines are NOAA Air Resources
Laboratory (ARL) Hybrid Single Particle Lagrangian Integrated Trajectory (HYSPLIT) 120 hour backward trajectories
initialized each hour of the day from BAO at 1000m above ground level (Stein et al., 2015). Trajectories were run using the
EDAS (Eta Data Assimilation System) 40 km x 40 km horizontal resolution reanalysis product (Kalnay et al., 1996). Figure
2 shows the smoke that impacted BAO during both periods was transported from large fire complexes in the Pacific
Northwest and Canada, with approximate transport timescales on the order of two to three days. Creamean et al. (2016)
provide a more detailed description of smoke transport and the sources of the aerosols associated with the August smoke-
impacted period.

## 172   4 Observed Changes in Ozone and its Precursors

### 173   4.1 CO, CH$_4$, and VOC Abundances

We quantified CO, CH$_4$, and 40+ VOC species including C$_2$-C$_{10}$ non-methane hydrocarbons (NMHCs), C$_1$-C$_2$ halocarbons,
and several oxygenated species (methyl ethyl ketone, acetone, and acetaldehyde) at BAO. The focus of the BAO field
intensive was to study the photochemistry of local emissions from oil and gas development (*e.g.* Gilman et al., 2013;
Swarthout et al., 2013; Thompson et al., 2014; Abeleira et al., in review), and we did not quantify species with known large
biomass burning emission ratios (*e.g.* hydrogen cyanide, acetonitrile, most oxygenated organic species). In addition, early
campaign issues with the online multichannel gas chromatography system compromised the data for the July smoke period
and thus we restrict our comparison of VOCs in smoke-free versus smoke-impacted periods to a comparison between 16 –
30 August, the *August smoke-impacted period*, and 24 July – 16 August, the *smoke-free period*. The brief smoke-free times
during 16 – 30 August (denoted by white between the red shading in Figure 1) were not included in either period since it is
difficult to determine whether they were smoke-impacted. GC measurements were made approximately every 50 minutes
and we compared 251 measurements of VOCs during the August smoke-period to 583 measurements during the smoke-free
period.

In this section, we describe significant changes in VOC abundances and notable exceptions. The HYSPLIT trajectories
(Figure 2) suggest that the age of the smoke impacting the Front Range during the August smoke-period was 2-3 days. We
observed enhancements in the abundances of CO, CH$_4$, and VOCs with lifetimes longer than the transport time of the smoke,
with the exception of some alkanes that have a large background concentration in the Front Range due to emissions from oil
and gas production. The alkenes we quantified (isoprene, ethene, and propene) were generally near the limit of detection





during the August smoke-impacted period, although notably cis-2-butene abundances were not changed. Significant
differences were not observed in the four oxygenated VOCs quantified between smoke-impacted and smoke-free periods.

Mean CO mixing ratios were significantly enhanced by 86 ppbv, or 65%, during the August smoke-impacted period (Figure
1). This enhancement was present across the diurnal cycle (Figure 3) and a greater range of CO mixing ratios (96 – 402 ppbv
versus 70 – 291 ppbv) were measured during the August smoke-impacted period compared to the smoke-free period.
Average enhancements of $CH_4$ were a much smaller percentage of (~3% or 67 ppbv), but comparable in magnitude to, the
CO mixing ratio enhancement. Methane has a relatively high background at BAO due to large emissions of $CH_4$ in nearby
Weld County from livestock production and oil and gas development (Pétron et al., 2014; Townsend-Small et al., 2016).
Taken together, the larger background of $CH_4$ and the large local sources of $CH_4$ in the Front Range served to mute the
impact of the August smoke on overall $CH_4$ abundances. The diurnal cycle of $CH_4$ did not change during the smoke-
impacted period as compared to the smoke-free period and we observed a similar range of mixing ratios (~1,840 – 3,360
ppbv) in the both smoke-free and smoke-impacted periods. We note several large spikes on the order of minutes during the
smoke-impacted period, but we do not believe that these are related to the presence of smoke because they were not
correlated with similar excursions in CO and PANs, and exhibited strong correlations with propane and other tracers of oil
and gas and other anthropogenic activity.

Similar to CO, ethane has an atmospheric lifetime on the order of a month during summertime at mid-latitudes (Rudolph and
Ehhalt, 1981) and is emitted by wildfires (Akagi et al., 2011). However, average ethane mixing ratios were not higher during
the August smoke-impacted period compared to the smoke-free period. One potential reason for this may be the large local
sources of alkanes from oil and natural gas activities within the Denver-Julesberg Basin which contribute to relatively high
local mixing ratios of these species (Gilman et al., 2013; Swarthout et al., 2013; Thompson et al., 2014; Abeleira et al.,
2017). The range of ethane mixing ratios observed at BAO was also not different between smoke-free (0.3 - 337 ppbv) and
smoke-impacted periods (1 – 362 ppbv), but the amplitude of the median diurnal cycle was dampened during the August
smoke-impacted period (not shown). Median morning ethane mixing ratios were ~10 – 20 ppbv less during smoke-impacted
than smoke-free periods, while afternoon and evening median mixing ratios were ~5 – 10 ppbv larger.  Most of the $C_3 – C_9$
alkanes we quantified showed similarly dampened amplitudes in their median smoke-period diurnal cycles. A consistently
lower planetary boundary layer (PBL) height during the day and a consistently higher boundary layer at night is one
potential explanation for these observations; however an estimate of the PBL height in the grid box surrounding BAO from
the North American Regional Reanalysis product (Mesinger et al., 2006) did not show any significant changes in PBL height
between the smoke-impacted and smoke-free periods. Likewise estimated PBL heights following methods from Coniglio et
al. (2013) and using atmospheric soundings at 0Z and 12Z in Denver (http://mesonet.agron.iastate.edu/archive/raob/) did not
show any differences between smoke-impacted and smoke-free periods (Figures S1 and S2). Figure 3 shows there were two
exceptions to the general alkane pattern noted above: 2-methylhexane showed a significant decrease in average abundances





(-39 pptv or -45%) and 3-methylhexane showed a significant increase (63 pptv or 75%) during the smoke-impacted period,
despite both having similar smoke-free abundances and similar rate constants for reaction with OH radicals ($\sim 7 \times 10^{12}$ cm$^3$
molec$^{-1}$ s$^{-1}$).

The atmospheric lifetimes of the four alkenes we quantified (isoprene, propene, ethene, and cis-2-butene) range from tens of
minutes to hours. Isoprene, propene, and ethene showed significant decreases in their average abundance:  -64% (-143 pptv),
-77% (-39 pptv), and -81% (-206 pptv) respectively. The shape of the diurnal cycles did not change (not shown), though
propene and ethene were near their respective limits of detection for the majority of each day during the smoke-impacted
period. These alkenes were among the most reactive species quantified, and one potential explanation for the reduced
abundance of these species during the smoke-impacted period is enhanced oxidation capacity linked to the presence of
smoke. However, we do not observe decreased abundances of cis-2-butene, which has a comparable OH-reactivity to
propene and lower average abundance. An alternative hypothesis for the reductions in the other three alkene species may be
reductions in local biogenic emissions during the smoke-impacted period either due to lower air temperatures or due to a
reduction in photosynthetic active radiation (PAR) at the surface during the August smoke-impacted period.  Isoprene is
emitted by broad leaf vegetation, and emission rates are highly light and temperature sensitive (Guenther et al., 2006).
However, while we did observe lower average daytime temperatures at BAO during the August smoke-impacted period
compared to the rest of the dataset (-2.3˚C), the majority of Front Range emissions of propene and ethene are likely from
anthropogenic sources. Thus this hypothesis could possibly help explain reduction in isoprene but not likely explain
reductions in ethene and propene. Shifts in local transport could also help explain differences but we did not observe any
consistent shifts in wind direction or changes in wind speed that would indicate consistently different local transport during
the August smoke-impacted period.

The only alkyne measured was ethyne. Ethyne is emitted by wildfires (Akagi et al., 2011) and has a lifetime of ~1 month
during summer. We observed a significant increase in the abundance of ethyne during the August smoke-impacted period.
These enhancements were small in absolute mixing ratio (0.163 ppbv), but represented a large percentage increase (67%)
and were consistently present throughout the day.

It is well known that wildfires produce carcinogenic aromatic hydrocarbons including benzene (Fent et al., 2014). During the
smoke-impacted periods, we observed significantly enhanced benzene throughout the day with an average increase of 0.117
ppbv and a percentage increase of 67%. These enhancements followed the pattern of CO and ethyne; there were consistent
increases throughout the day and the diurnal cycle retained its shape. Wildfires also produce toluene (Fent et al., 2014);
however, it has a substantially shorter lifetime (< 2 days) than benzene (~12 days). Toluene showed no significant changes in
its mean mixing ratio, diurnal cycle, or range of values measured at BAO during the smoke-impacted periods. The other
aromatic hydrocarbons we quantified (o-xylene and ethyl-benzene) also did not change significantly.






As mentioned in Section 1, oxygenated VOCs are emitted by wildfires and make a large contribution to the total emitted
VOC mass in wildfire smoke (Stockwell et al., 2015). Additionally they are produced as oxidation intermediates (Atkinson
and Arey, 2003). Acetaldehyde, acetone, and methyl ethyl ketone (MEK) showed no consistent changes in their abundances,
diurnal cycles, or range during the smoke-impacted period compared to the smoke-free period. Small increases in average
acetone (~350 pptv) and MEK (~150 pptv) mixing ratios during late afternoon and evening hours were not statistically
significant.

Given the diversity of emission sources across the northern Colorado Front Range, previous studies of atmospheric
composition at BAO have noted a strong dependence of VOC composition on wind direction (Pétron et al., 2012; Gilman et
al., 2013). Recent housing development and oil and gas production surrounding the BAO site have made analyses based on
wind direction more challenging in recent years (McDuffie et al., 2016). Importantly for our analysis, we found that the
statistically significant changes in all species during the smoke-impacted periods occurred across all wind directions. Figure
4 shows this for two example species: benzene and $NO_2$. We also did not find statistically significant changes in wind
direction or wind speed patterns between smoke-free and smoke-impacted periods. Thus we attribute the changes in
atmospheric composition during the August smoke-impacted period to the presence of smoke.
**4.2 Reactive Oxidized Nitrogen ($NO_y$) Species**
Peroxyacyl nitrates and $HNO_3$ were successfully measured from 10 July – 7 September and alkyl nitrates were measured
from 24 July – 30 August. Thus we report significant changes in these species for the August smoke-impacted period only.
We observed significant enhancements in both peroxyacetyl nitrate (PAN) and peroxypropionyl nitrate (PPN) during the
August smoke-impacted period. PAN and PPN abundances were consistently elevated across the day by an average of 183
and 22 pptv respectively, corresponding to a ~100% change for both species. The peak of each diurnal cycle was shifted later
in the day by about 3-4 hours for the smoke-impacted period. This cannot be accounted for merely by the shift in the timing
of solar noon given that the total decrease in daylight between 10 July and 30 August is ~2 hours. The ratio of PPN to PAN
during the August smoke-impacted period exhibited a significant decrease from the smoke-free period ratio ($0.14 \pm 0.012$
versus $0.17 \pm 0.006$; calculated as the slope of a reduced major axis linear regression on the hourly data from 12PM – 5PM
MDT Figure S3). The direction of change in the ratio is consistent with observations of PPN/PAN ratios in Asian urban and
aged biomass burning plumes off the coast of California (Roberts et al., 2004). The $C_1 – C_2$ alkyl nitrates measured at BAO
exhibited similar behaviors; methyl nitrate and ethyl nitrate saw average enhancements during the August smoke period of
1.2 and 0.77 pptv, 41% and 31% respectively, though the average mixing ratios of these species are smaller by an order of
magnitude compared to other alkyl nitrates quantified. Propyl-, pentyl-, and butyl-nitrate did not display significant changes
in their average mixing ratio, though we observed a similar shift in the peak of their diurnal cycles of 2-4 hours. We did not
observe significant changes in the abundances of $HNO_3$. There were no changes to the diurnal cycle or the range of mixing





ratios observed.

NO and $NO_2$ measurements were made during the entire campaign, 1 July – 7 September 2015, so both the July and August
smoke-impacted periods were analyzed with respect to potential changes in $NO_x$. NO was present in the same abundances
between the two periods and showed the same diurnal cycle during the August smoke-impacted period as compared to the
smoke-free period (Figure 5). During the July smoke-impacted period the morning buildup of NO was slower than the
smoke-free period, though the mixing ratios were within the range of smoke-free values and there were fewer days in the
July period compared to the August smoke-impacted period.

Figure 5 shows that $NO_2$ abundances exhibited more significant changes. During the July smoke-impacted period, $NO_2$ was
within the range of smoke-free measurements but the diurnal cycle was shifted later in the day and the average decrease in
mixing ratios of $NO_2$ in the afternoon was not as strong as during the smoke-free periods. In contrast $NO_2$ during the August
smoke-impacted period followed the same diurnal cycle but had pronounced significant increases in average mixing ratios
during the morning and evening hours of ~8 ppbv (17%) following sunrise and 3 ppbv (60%) following sunset. These
enhanced peak abundances appeared during multiple days during the August smoke-impacted period. We did not find
evidence that these enhancements were due to traffic patterns. The concurrently observed PAN abundances can only account
for at most 1 ppbv of additional $NO_2$, but there could have been significantly higher PAN abundances in the smoke plume
prior to reaching BAO and PAN dissociation is one hypothesis for the enhanced abundances. We do not have measurements
of other reactive nitrogen species (e.g. HONO, $ClNO_2$, $NO_3$, and $N_2O_5$) to test potential other hypotheses of a different
chemical mechanism to explain the observed $NO_2$ enhancements.

### 4.3 Ozone

As discussed in the introduction, wildfire smoke has been found to produce $O_3$ within plumes and to be correlated with
enhanced surface $O_3$ in areas to which it is advected. The total amount of $O_3$ at a location is a complex combination of the
relative abundances of VOCs and $NO_x$, meteorological conditions supporting local $O_3$ production, and the amount of $O_3$
present in the air mass before local production. In this section we describe the significant increases in $O_3$ during both smoke-
impacted periods, show that these enhancements were most likely not due to changes in meteorological conditions, and
discuss evidence pointing to whether these changes may be due to enhanced local production or transport of $O_3$ produced
within the smoke plume.

Figure 5d shows that there were significant increases in $O_3$ mixing ratios during nighttime and midday during the August
smoke-impacted period compared to the average smoke-free diurnal cycle. The mean $O_3$ mixing ratio across all hours of the
day was 6 ppbv (14%) larger during the August smoke-impacted period than the smoke-free period (Figure 6), significant at





the 99% confidence level based on a two-sample difference of means t-test. There were no significant changes in the average
$O_3$ mixing ratios during the July smoke-impacted period (Figure 5a). The average mixing ratio of $O_3$ during the July smoke-
impacted period was not greater than absolute average during the smoke-free period (Figure 5a). However, as discussed in
Section 2, this period in particular was much colder on average than the smoke-free period.

$O_3$ mixing ratios generally increase with temperature, and this relationship has been attributed to 1) warm and often stagnant
anti-cyclonic atmospheric conditions that are conducive to $O_3$ formation, 2) warmer air temperatures that reduce the lifetime
of PAN, releasing $NO_2$, and 3) lower relative humidity that reduces the speed of termination reactions to the $O_3$ production
cycle (Jacob et al., 1993; Camalier et al., 2007). Figure 6 presents hourly average $O_3$ and temperature at BAO and shows a
positive relationship between $O_3$ and temperature for both the smoke-free period in black and August smoke-impacted period
in red. The increase in $O_3$ mixing ratios during the August smoke-impacted period compared to the smoke-free period is
present across the entire range of comparable temperatures. Figure S4 shows the same result during the July smoke-period,
where for comparable temperatures the July smoke-period has higher $O_3$ than would be expected from the $O_3$-temperature
relationship during the smoke-free period.  Across both smoke-impacted periods and for a given temperature, the magnitude
of the increase in average $O_3$ was $10 \pm 2$ ppbv. This was calculated as the mean difference between medians within each
temperature bin weighted by the total number of hourly measurements within each bin. The weighted standard deviation was
calculated in the same way. The magnitude of this difference is greater than the average difference in means between the
smoke-free $O_3$ mixing ratios and the August smoke-impacted period because there were several periods during the July and
August smoke-impacted period where air temperatures were colder ($\sim 5°C$) than most observations during the smoke-free
period. Thus the lower $O_3$ mixing ratios associated with these smoke-impacted periods (*e.g.* $\sim$ 20 - 40 ppbv) were not
included in the weighted difference in medians since there were not commensurate smoke-free $O_3$ measurements at those
same temperatures.

In addition to a positive relationship with surface temperature, elevated $O_3$ in the western U.S. has also been found to be
correlated with 500 hPa geopotential heights, 700 hPa temperatures, and wind speeds (Reddy and Pfister, 2016). We tested
the relationship between $O_3$ and these meteorological variables during our study period using observations from the 0Z and
12Z atmospheric soundings conducted in Denver (http://mesonet.agron.iastate.edu/archive/raob/). We did not find a
relationship between $O_3$ and daily 500 hPa geopotential heights or 700 hPa temperatures, nor were these meteorological
variables notably elevated during the August smoke-impacted period (Figures S5, S6, and S7). Additionally we did not find
a significant change in wind speed during the August smoke-impacted period. Thus we have no evidence that the enhanced
$O_3$ during the August smoke-impacted period was due to meteorological factors.

To determine if a change in synoptic scale transport in smoke-impacted versus smoke-free periods could have contributed to
different abundances, we performed a k-means cluster analysis on 72-hour HYSPLIT back trajectories. The trajectories were



calculated using the methods described above, and initiated each hour at 2000 m a.g.l. from BAO. We chose to initialize the
trajectories at 2000 m a.g.l so that fewer trajectories intersect the ground in the Rocky Mountains.  Trajectories are unlikely
to capture the complex circulations (*e.g.* potential Denver Cyclones or up/down slope winds) characteristic of summertime in
the Front Range, but they should capture synoptic scale air mass motions. The k-means analysis clustered each trajectory
into a predetermined number of clusters by minimizing the distance between each trajectory and its nearest neighbor; this
technique has been used to classify air mass history in air quality studies (Moody et al., 1998). We found 4 predominate
trajectory clusters during our study period: northwesterly flow, westerly flow, southwesterly flow, and local/indeterminate
flow (Figure S8). We then compared afternoon (12PM – 5PM MDT) hourly $O_3$ measurements separated by trajectory cluster
and binned by temperature between the smoke-free period and the August smoke-impacted period. Most hours during the
August smoke-impacted period were associated with northwesterly flow and we found the same enhancement in $O_3$ for a
given temperature when comparing smoke-impacted observations to smoke-free observations assigned to this cluster as we
found for the complete dataset (Figures S9 and S10). Thus we conclude that potential changes in $O_3$ driven by synoptic scale
transport conditions cannot account for the observed $O_3$ enhancements during the August smoke-impacted period at BAO.

We calculated the maximum daily 8 hour average (MDA8) $O_3$ mixing ratios, following methodology from the U.S. EPA,
and found that out of 6 high $O_3$ days at BAO (defined as > 65 ppbv MDA8) during our study period, 2 occurred during the
August smoke-impacted period (Figure 7). As we stated above, elevated $O_3$ during the smoke period was not a result of
abnormal meteorological variables such as higher than normal temperatures, and thus these 2 high $O_3$ days are very likely
caused in part due to the presence of wildfire smoke. The lower portion of Figure 7 again shows that maximum daily
temperatures during the smoke-impacted periods were the same as or lower than maximum daily temperatures during the
smoke-free period.

To assess the spatial extent of the $O_3$ enhancements observed at BAO and to investigate the relative likelihood of $O_3$
enhancements due to transport within the smoke versus greater local production, we analyzed hourly $O_3$ measurements from
two nearby National Park Service (NPS) Air Resources Division (http://ard-request.air-resource.com/data.aspx)
measurement locations. The Rocky Mountain National Park long-term monitoring site (ROMO; 40.2778°N, 105.5453°W,
2743 meters A.S.L.) is located on the east side of the Continental Divide and co-located with the Interagency Monitoring of
Protected Visual Environments (IMPROVE) and EPA Clean Air Status and Trends Network (CASTNet) monitoring sites.
Front Range air masses frequently reach this site during summer afternoons (Benedict et al., 2013). The Arapahoe National
Wildlife Refuge long-term monitoring site (WALD; 40.8822°N, 106.3061°W, 2417 meters A.S.L.) near Walden, Colorado,
is a rural mountain valley site with very little influence from anthropogenic emissions. Figure 8 shows that the August
smoke-impacted period produced increases in $O_3$ mixing ratios across all three sites.
When comparing afternoon data for a given temperature, there are enhancements of 10 ± 2 ppbv, 12 ± 3 ppbv, and 4 ± 2
ppbv $O_3$ at BAO, ROMO and WALD respectively. $O_3$ enhancements across all three sites, across an approximate urban to





rural gradient, suggest that some amount of the $O_3$ enhancement observed at BAO during the August smoke-impacted period
is the result of $O_3$ production within the plume during transit. As for enhanced local production of $O_3$, we do not find any
significant differences in average calculated ozone production efficiency (OPE) between the smoke-impacted and smoke-
free periods (Figure S11) but the VOC composition changes suggest possible enhanced oxidation capacity during the August
smoke period. Fully addressing the question of whether the smoke enhanced local $O_3$ production in the polluted Front Range
requires the use of a chemical transport model, and is beyond the scope of this work.
**5 Conclusions**
Here we report a time series of detailed gas-phase ground measurements in the northern Colorado Front Range during
summer 2015. Clear anomalies in CO and $PM_{2.5}$ showed that aged wildfire smoke was present at ground-level during two
distinct periods (6 – 10 July and 16 – 30 August) for a total of nearly three out of the nine weeks sampled. This smoke from
wildfires in the Pacific Northwest and Canada impacted a large area across much of the central and western U.S., and was
several days old when it was sampled in Colorado. This wildfire smoke mixed with anthropogenic emissions in the Front
Range, resulting in significant changes in the abundances of $O_3$ and many of its precursor species. Our measurements are
unique because of 1) the length of time we sampled this smoke-impacted anthropogenic air mass, and 2) the detailed
composition information that was collected.

During the smoke-impacted periods we observed significantly increased abundances of CO, $CH_4$, and several VOCs with
OH oxidation lifetimes longer than the transport time of the smoke. We measured significant decreases in several of the most
reactive alkene species, indicating possible enhanced oxidation processes occurring locally. Mixing ratios of peroxyacyl
nitrates and some alkyl nitrates were enhanced and peak abundances were delayed by 3-4 hours, but there was no significant
change in $HNO_3$ mixing ratios or its diurnal cycle. During the longer August smoke-impacted period we observed significant
increases in $NO_2$ mixing ratios just after sunrise and sunset. We did not observe any consistent shifts in wind direction or
changes in wind speed that can explain the observed changes in composition, and the changes in abundances that we
observed for a given species were generally present across all directions and speeds.

We observed significantly enhanced $O_3$ abundances of about 10 ppbv for any given temperature during both smoke-impacted
periods. The enhancements during the August smoke-period led to very high surface $O_3$ levels; out of 6 high $O_3$ days at BAO
during our study period, 2 were during the August smoke-period and were impacted by wildfire smoke. These enhancements
were not due to higher temperatures, nor anomalous meteorological conditions. We found evidence of $O_3$ produced within
the smoke plume during transit, with potentially enhanced local production as well due to enhanced oxidation capacity.





It is important to note that the presence of smoke does not always result in very high $O_3$ abundances. Many other factors
contribute to the overall level of surface $O_3$, and smoke can also be associated with decreased $O_3$ at times, such as during the
July smoke event described herein. Each smoke event has unique characteristics and thus it is important to study and
characterize more events such as these in the future.

Wildfire smoke during these time periods most likely impacted atmospheric composition and photochemistry across much of
the mountain west and great plains regions of the U.S. Given the BAO, Rocky Mountain and Walden research locations span
an urban-rural gradient as well as a large altitudinal gradient, it is likely that both rural and urban locations impacted by this
smoke could have experienced enhanced $O_3$ levels. Wildfires are increasing in both frequency and intensity throughout the
western U.S. due to climate change and thus wildfire smoke events such as this one will likely play an increasingly prevalent
role in degrading U.S. air quality.

**Author Contribution:** J. L. compiled and analysed the data, and wrote the manuscript. All authors participated in data
collection at BAO and contributed to the writing of or provided comments on the manuscript.

**Acknowledgements:** Funding for this work was provided by the US National Oceanic and Atmospheric Administration
(NOAA) under Award number NA14OAR4310148. Support for Jakob Lindaas was provided by the American
Meteorological Society Graduate Fellowship. We appreciate all the logistical help at BAO provided by Dan Wolfe, Gerd
Hübler, and Bruce Bartram. We appreciate access to NOAA GMD ozone data provided by Audra McClure-Begley. Thank
you also to Jake Zaragoza and Steven Brey for assistance at BAO and for running the HYSPLIT trajectories.

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




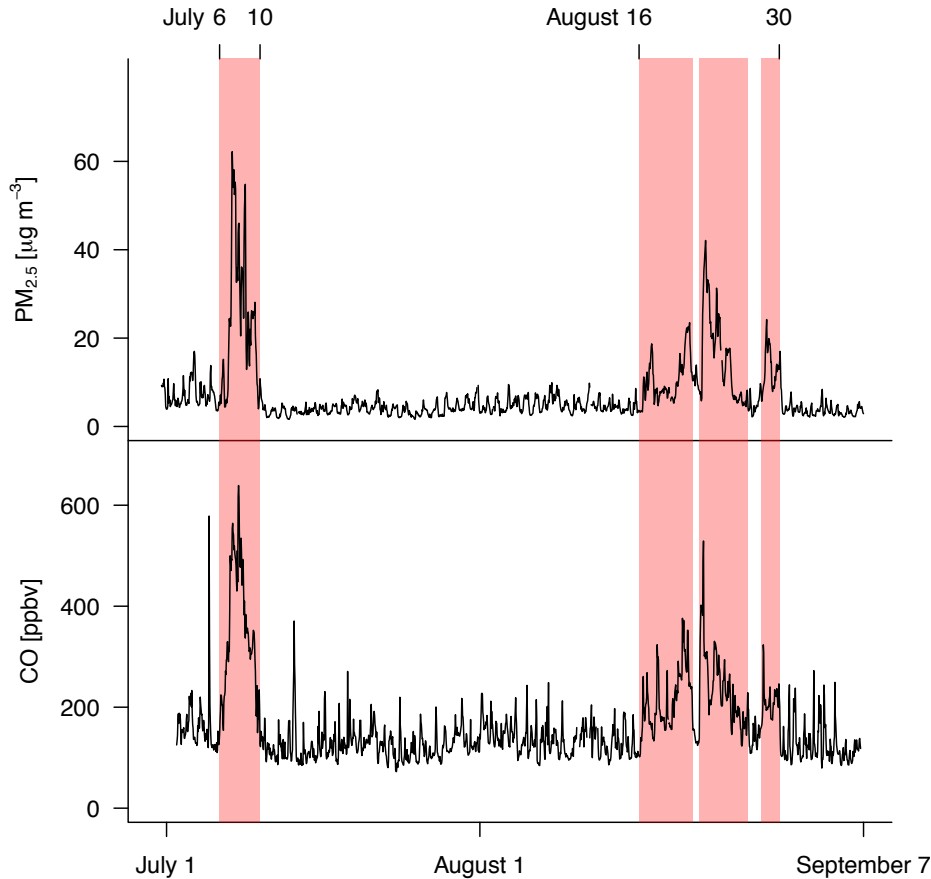


**Figure 1. Top panel: Time series of hourly PM$_{2.5}$ concentrations for the CDPHE CAMP air quality monitoring site (www.epa.gov/airdata) located in downtown Denver (39.75', -104.98'). Bottom panel: Time series of hourly CO mixing ratios at the Boulder Atmospheric Observatory (BAO: 40.05', -105.01'). Red shading denotes periods during which smoke is present at BAO.**





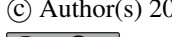

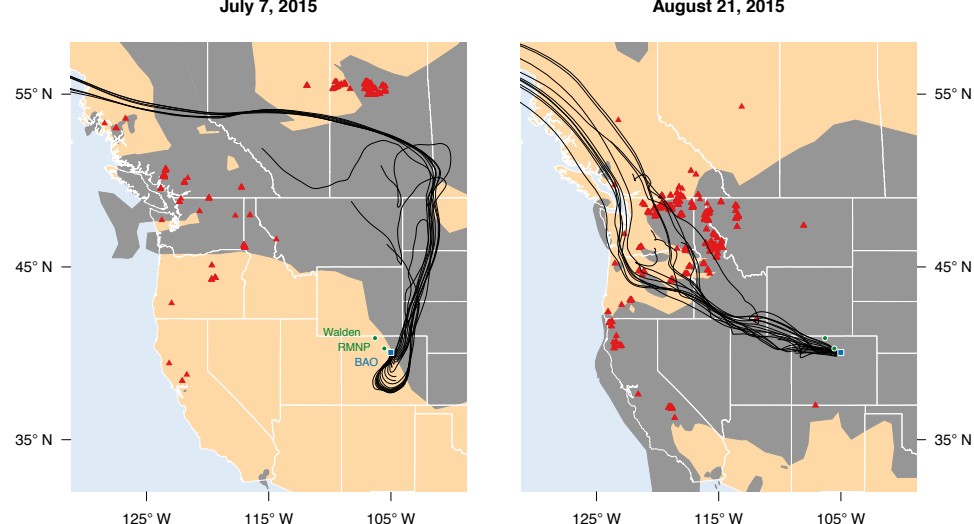


**Figure 2. Representative days during each smoke period observed at the Boulder Atmospheric Observatory (BAO: blue square).**
**NOAA Hazard Mapping System (http://www.ssd.noaa.gov/PS/FIRE/) smoke polygons are plotted in grey with MODIS fire**
**locations (http://modis-fire.umd.edu/index.php) from the previous day plotted as red triangles. The thin black lines show**
**HYSPLIT back trajectories from the BAO site initiated 1000 m a.g.l. for each hour of the day plotted.**





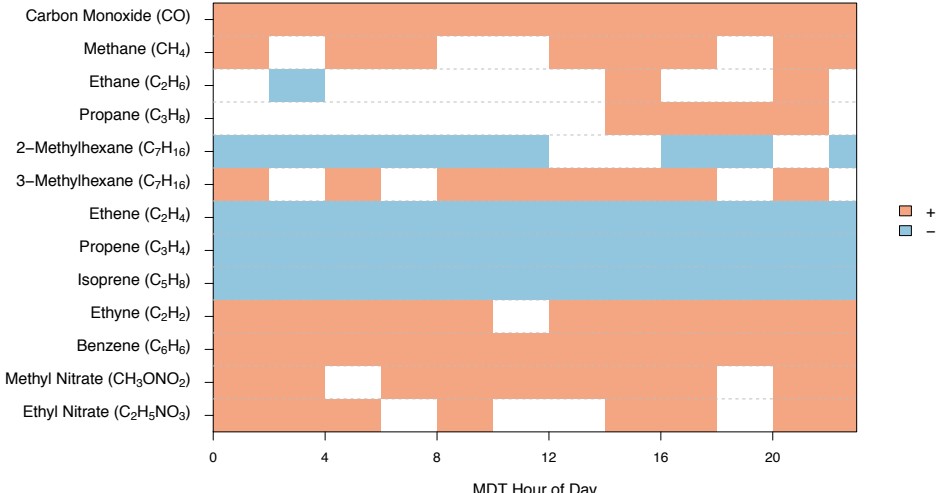


**Figure 3. Significant changes (two sided Student's t-test, 90% confidence interval) in hourly averaged mixing ratios of a subset of**
**species measured at BAO between smoke-free periods and the 16 - 30 August smoke period. Significant increases during smoke-**
**impacted periods compared to smoke-free periods are shown in red, significant decreases are in blue.**

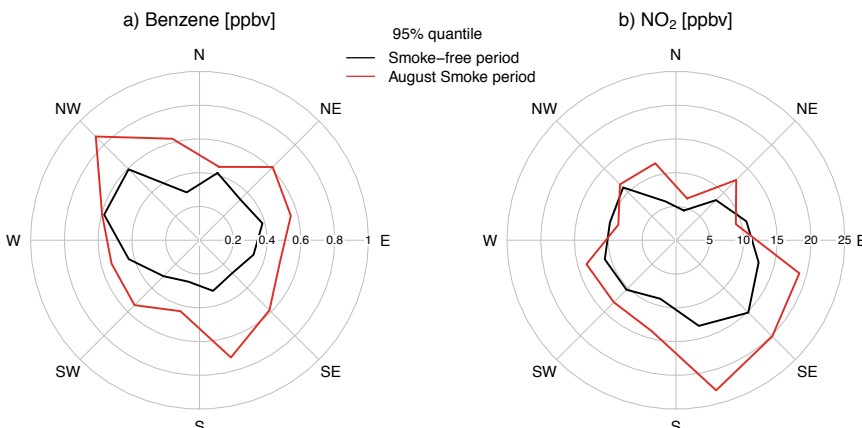


**Figure 4. 95th percentiles of a) benzene and b) NO$_2$ as a function of wind direction for all data during smoke-free periods (black)**
**and the August smoke period (red).**





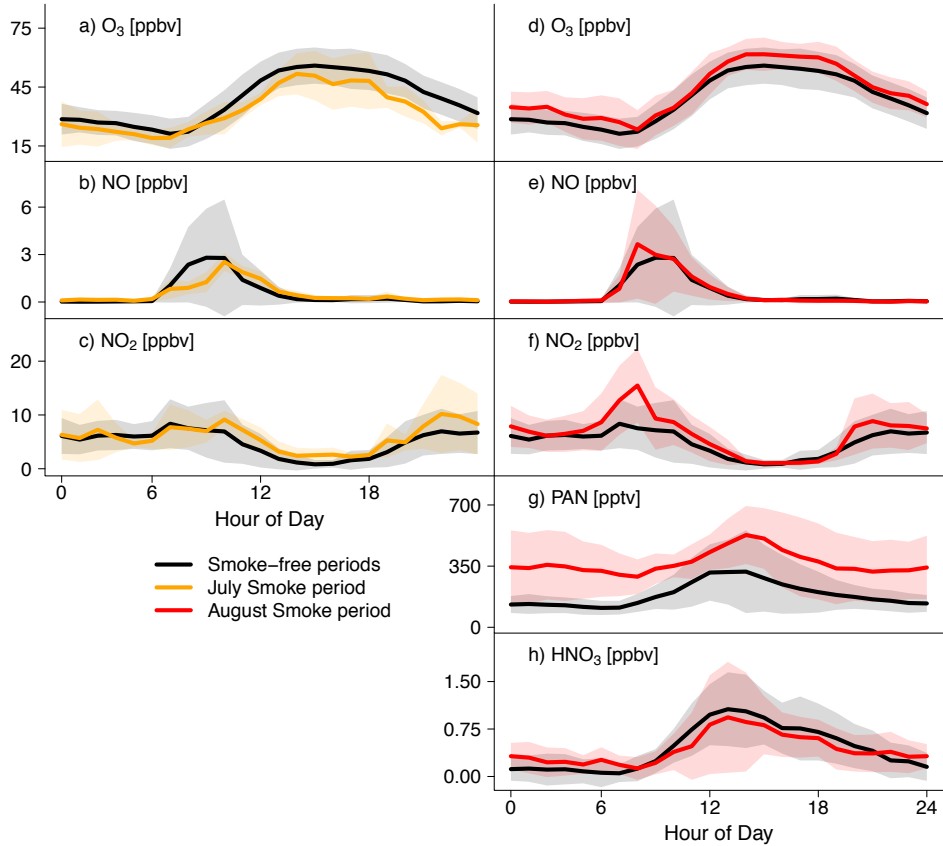


**Figure 5. Average diurnal cycles in MDT of O₃ and oxidized reactive nitrogen species at BAO. Panels a), b), and c) compare**
**average diurnal cycles from smoke-free time periods (black) to average diurnal cycles from the July smoke-impacted period**
**(orange). Panels d) – h) show average diurnal cycles during the August smoke-impacted period (red) to the same average diurnal**
**cycles from smoke-free periods (black). PAN and HNO₃ measurements were not available during the July smoke-impacted period.**





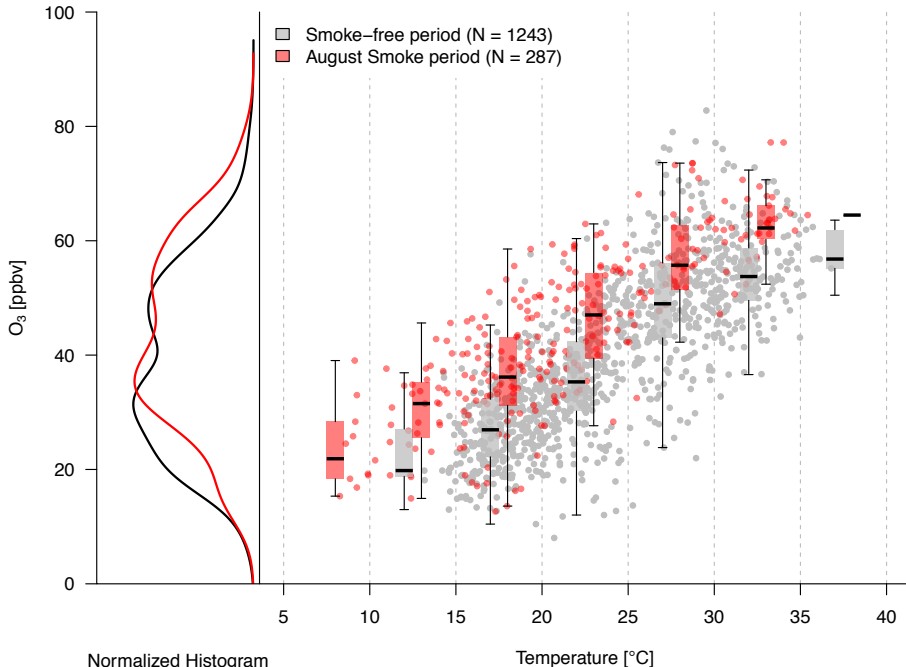

669

**Figure 6. Hourly O$_3$ data from BAO plotted against hourly temperature data show a positive correlation between temperature and O$_3$ abundances for both smoke-free time periods in grey and the August smoke-impacted time period in red. Overlaid are boxplots (5th, 25th, 50th, 75th, and 95th percentiles) for each 5 °C bin. On the left normalized histograms of the hourly O$_3$ data are plotted, with all smoke-free measurements in black, and all hourly measurements made during the August smoke-impacted period in red.**



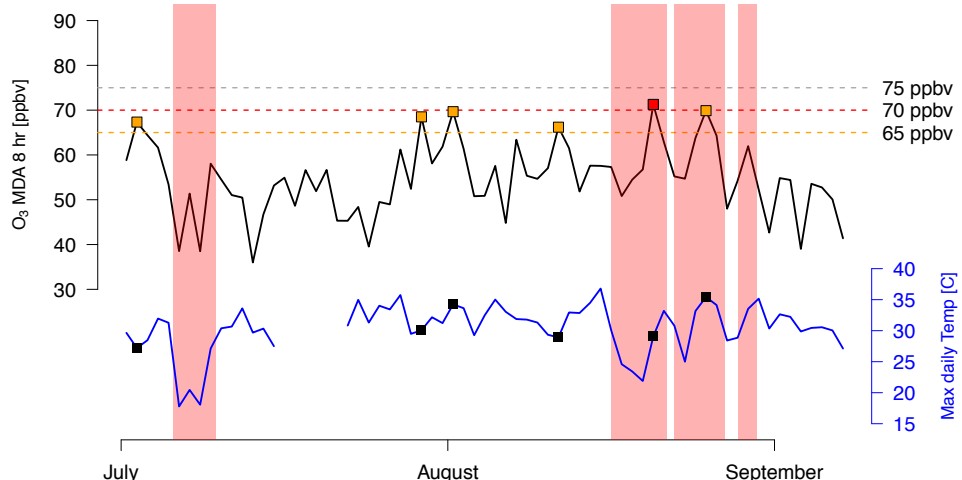

674

**Figure 7. Maximum daily 8-hour average (MDA8) O₃ mixing ratios at BAO plotted in black with maximum daily temperature at BAO in blue. Orange and red boxes denote days that exceed 65 and 70 ppbv respectively.**





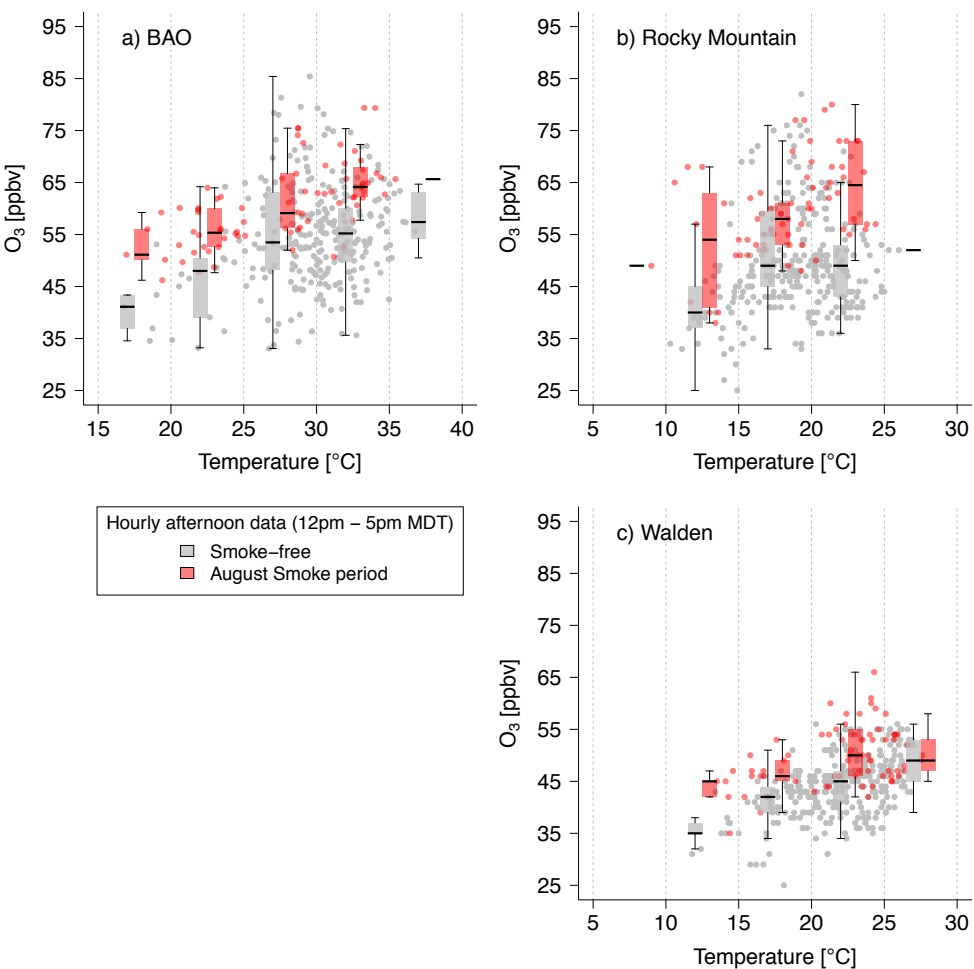

677

**Figure 8. Hourly O$_3$ versus temperature for a) BAO, b) the Rocky Mountain National Park long-term monitoring site, and c) the Arapahoe National Wildlife Refuge long-term monitoring site near Walden, CO. Plotted here are hourly afternoon data (12PM – 5PM MDT), with boxplots showing standard percentiles of 5 °C binned O$_3$ data the same as was shown in Figure 6.**