# Peer review of "Changes in ozone and precursors during two aged wildfire smoke events in the Colorado Front Range in summer 2015"

_Atmospheric Chemistry and Physics, 2017_

## Referee Comment (RC1) · Anonymous Referee #1 · 18 Apr 2017

MS No.: acp-2017-171

General Comments

Overall, this is a good example of a case study, with relevance specifically to Colorado air quality and which, more generally, speaks to air quality concerns relevant to the western United States, where wildfires are a substantive air quality issue. In Colorado, there is an active community of regulators at the State and Federal level who have been debating the very issues discussed in this paper for well over a decade. There is an extensive network of monitoring and also substantive photochemical modeling

address this issue from a policy perspective. This case study has merit, but the danger here is that a paper will over generalize a case study and overstate its own importance. The authors should be careful in this regard. From a regulatory perspective, actual exceedances of ambient standards for ozone (70 ppbv for 8 hours) are relevant and anything below this is generally not relevant. Even so, a non-attainment designation is based on more than a single exceedance at a single location. The authors should demonstrate that knowledge and perspective in the body of this work. It is well established that wildfire smoke can enhance ozone formation, especially air masses that have been aged for several days. Despite an extensive monitoring network and a concentrated field study, measurements are sparse, as is evidenced here by the use of data from several monitors remote from the BAO tower. This paper would be strengthened immensely by the use of satellite imagery showing the wildfire smoke haze during the periods of interest and also by the use of additional State and Federal agency data to establish that wildfire smoke was the cause of the ozone enhancements observed. In my mind, this is not unequivocally established by the observations presented here. For example, could CO enhancements be caused by Asian airmass transport? I encourage the authors of this work make substantial improvements to this work before I recommend publication of this work in Atmospheric Chemistry and Physics. Also, in this work, the authors contrast data from two "smoke" events with that from non-impacted periods during the same period. However, much of the data from the first fire plume is disregarded. It's excluded from much of the text and the figures. In my opinion, the first fire plume should be included in all analyses, even if the results are diminished. If this is really a fire plume, it should not be dismissed and excluded. Fire plumes are variable, and that is an important point. Sometimes they make a case easier, sometimes more difficult, but this is a reality in a complex world.

Specific Comments

1) The Title and Abstract should more strongly indicate that this is a single case study shoing influences from remote wildfire smoke on one location downwind

2) The Abstract should mention if nearby official monitors showed ozone exceedances to put this case study into context (see additional comments to this effect below).

3) Introduction. I recommend that the authors extend the background discussion to include policy relevant discussions and demonstrate knowledge of the extensive regulatory work that is ongoing on this topic in the west related to ozone exceedances. For example, only one exceedance of the NAAQS for ozone occurred for all of July and August at the measurement site. How does this compare to the exceedences for the entire State for 2015? Was the entire State in non-attainment this year? Was the event mentioned here a contributing factor? Or, did it fall much farther down the list other exceedences of the standard in the Front Range in 2015. These questions are very relevant to policy and should be discussed in some detail to place the study in a larger context.

4) Introduction/Smoke events: How does the climatology of the study period compare to Colorado's as a whole? Was it a cool year? A hot year? A wet year? Was there evidence of pollution transport besides fire smoke from other areas of the US and Internationally?

5) It is not evident why values from the study site were compared with very select other monitoring sites (in this case CAMP, ROMO and Walden) and not others. This gives the impression that supporting evidence has been picked rather than evaluated broadly. Why, why for example, is the PM from the CAMP site (20+ miles from the BAO tower) used, while the CO measurements from CAMP have not? Why are other Front Range ozone measurements not evaluated? Why were Walden and Rocky Mountain ozone sites used, while other data (e.g. CO data from the Storm Peak Lab) were not? This needs to be address directly in the text.

6) Was there satellite imagery from this period that showed the wildfire smoke haze? This data is often widely available, and so should be included if possible.

7) Line 102. The GC-method needs to be summarized in more detail. What is the

integration period? What is the frequency of measurement? The overall method can be referenced from another paper, but those parameters are important and should be included here.

8) Line 147 and Figure 1. CO and PM2.5 data from other surrounding monitors should also be included. Ozone data from other Front Range Non-attainment area monitors should be summarized and discussed.

9) Figure 1. Add CO from CAMP at minimum. There are other CO monitors nearby, do they agree? Add PM2.5 from locations closer than CAMP if possible. Address the latter points in text if they cannot be added to the figure.

10) Lines 147-157. CAMP also has O3. The data from that station's O3 needs to be included/shown here, and any place PM2.5 is used from that site, especially given the 35 km distance between sites. Also, there needs to be a space between 35 and km on line 150. The paper states, "PM2.5 was similarly elevated during the smoke-impacted periods at CDPHE monitoring sites across the Colorado Front Range (not shown)." Why is it not shown? It should be. Lastly, the spikes in figure 1 data are of equal magnitude as the spikes within the defined periods, why are these smoke free?

11) Figure 2. Recommend that satellite imagery of smoke added as additional figure to make the case that the plume was smoke and widespread.

12) Section 4.1. Measured data, especially for VOCs, should be tabulated and summarized. Please insert a relevant table of species measured with relevant max, min, median values and standard deviations.

13) Figure 3. Labels not clear. Add text labels. Why is first fire period excluded?

14) Figure 3 is hard to understand without a table or plot of VOC concentrations.

15) Lines 178-181. The fact no biomass burning specific VOC speciation was done at all seems a bit off. This is surprising given the title, and conclusions, and likely impacts of this paper.

16) Line 188. Section 4.1, Figure 2: Unclear how the statement "... suggests that the age of smoke impacting the Front Range during the August smoke-period was 2-3 days." That is not apparent in the figure.

17) Figure S1-S2. These plots are jumbled. Add legends. If your point is that the boundary layers at 0Z are more variable than the ones at 12Z, you should make that point in the text. The data contradict the conjecture you make around line 218. It's not clear why the sondes are included here. The surface temperature data is presented in Figure 8, so why show the sondes? Perhaps these figures should be revised to be simpler and more concise, or removed. If you must show the soundings then perhaps have two panels, one for smoke free and one for smoke and then a solid gray-area representing all the data, and a line for the average, or even a vertical box/whisker plot.

18) Line 216. "Not shown" in reference to the diurnal cycles. Diurnal cycles should be shown.

19) Line 218. A lower PBL height during the day is exactly the opposite of what is observed and this directly contradicts data from figures S1 and S2. This speculation should be removed.

20) Line 231-2. This statement does not make sense. Abundances decrease over what time period? Please clarify the wording.

21) Line 232. Diurnals not shown. Conclusions in this section could use the support of the diurnal cycles and as is it's hard to follow without them.

22) Paragraph including lines 230-246. Trends are not explained and speculation here is spurious. A table presenting the measured values could easily replace this table. Also, why would isoprene behave differently than other alkenes? Are the changes in alkenes even significant given that they are near their detection limits? A table would suffice here rather than trying to explain trends in ways that mean little. It is unclear what the conclusion of this paragraph is. It's also unclear what the take home point is

or and how the evidence supports the conclusion.

23) Figure 4. Include first fire period. This figure does not appear to be referred to in the text? It is unclear what 95th percentiles mean. In the legend says quantile and not percentile. Clarify. If it is not referred to in the text, it should be eliminated.

24) Figure 5. Indicate what shaded regions are. Are they percentiles? Of which measurements? Note that almost never does red line leave the grey shaded area, except for PAN and NOx. Discuss in text. Show solar noon on the plots for clarity.

25) Line 308. Please include more detail about the analysis you did related to traffic impacts.

26) O3 does indeed have a positive correlation with surface temperature as referenced often in this paper. However for the Front Range region, this should be tempered by the fact that the almost parallel rise in temperature and ozone starts dropping off after the air temperature hits about 86-90F (30C-32C). Some evidence of this can be seen in Figure 6. The reason for this is that once surface temperatures begin to exceed this threshold, a westerly wind component usually becomes dominant. These westerlies will often be gustier and not allow the cyclical terrain-driven circulations that normally enhance ozone concentrations across the Front Range. As referred to in this paper, the Reddy-Pfister study of 2016 expands on this and concludes that 500 mb heights and 700 mb winds hold a stronger correlation to ozone concentrations than surface temperature for Front Range locations. Perhaps this is irrelevant since the air temperature during the "smoke" periods did not get very hot, but maybe an explanation of this phenomena should be included if surface temperature is being emphasized as being more important than the other variables mentioned above.

27) Figure 6. Include first smoke impacted period on this chart in another set of colored box and whiskers if they are different from the second smoke period. Also, in this figure, the gray bars are indistinguishable from the gray circles, they blend together. Perhaps use black bars instead of gray. The same is true for Fig 8, S4, and S9.

28) Figure 7. The point highlighted in mid-August where temperature is low and O3 is high looks interesting, why is this not considered smoke influenced given the paper's hypothesis?

29) Figure S3. Clarify if the data shown is for one or both smoke free periods. Show both, using different are lines, if they are different from each other. It is hard to see what's happening at lower values due to so many points. Or figure could be revamped showing quantiles with error bars and all data in gray behind.

30) Figure S4. Combine this figure with Figure 6.

31) Line 323. Figure 5d doesn't appear to show a very significant difference in ozone between the black and red shaded areas. Perhaps, the figure needs to be edited to make the true difference clearer; otherwise, it seems overstated in the text.

32) Line 330-333. O3 production with temperature levels off at high temperatures particularly in the Front Range due to the wind speed and direction associated with these high of temperatures. This should be addressed in the text.

33) Lines 334-335. Things like black lines or red lines descriptions should be in the figure legend and caption, not text body.

34) Line 361-363. The chosen altitude limit makes sense, but the Denver cyclone and in-basin wind patterns do contribute to ozone and recirculation. This should be emphasized more and discussed. The authors should include the wind field reanalysis data to show surface winds on their chosen day of interest in each smoke period.

35) Line 340. How did the weighting occur? Insert a reference or elaborate.

36) Does the "synoptic scale transport" discussed at the end of page 11 and start of page 12 also account for the possibility of Asian pollution influence? The HYSPLIT back trajectories on page 20 both suggest that at least a portion of the air mass may have originated in Asia. It would be interesting to see just how much, if any, influence Asian pollution may have had when comparing the smoke and non-smoke air masses.

37) Line 352. It is unclear how just referencing the geopotential height paper (include citation at this location) leads to the conclusion that was "no evidence" of meteorological factors in ozone enhancement. This is a very broad generalization and needs supporting evidence and specific discussion if it is to be included here. Is the point you are making that the lack of meteorological factors that correlate with ozone implies that all the ozone was due to fire? If so, make this case strongly and state it clearly. Is absence of evidence meteorological driven ozone production even acceptable evidence? I'm not so sure it is. At best, it is supporting evidence.

38) Figure S5-7. The authors should explicitly discuss how the data in these figures supports their argument. This is a good supporting point, but there is need to flush out the discussion and figures better. Devoting 3 figures vague scatter plots to this is excessive. Could they be layered in 3 dimensions on a single plot? Alternatively, make one 3-panel figure or remove entirely and only quote the R2.

39) Line 368. Is this flow discussion where you should refer to Figure 4?

40) Figure 7. This figure needs to be put into context. Did the Front Range exceed the NAAQS this year? Was this one of the four maximum values that put the region into non-attainment for the year? Or was it much further down the list? This is valuable context information that should be discussed in the text. One exceedence is generally irrelevant to the overall policy discussion, but if this is not the case, it is certainly worth discussing in more detail.

41) Paragraph lines 373-375. Why did you pick 65 ppbv MDA8, when this is not the standard? This seems arbitrary. Please use the current standard and put into the correct context of this year's ozone for the entire area as mentioned in a previous comment. Please also adjust your conclusions accordingly.

42) Figure 8. Same comment as for Figure 6. This needs to include the first fire period. Also, why were these sites chosen? Is it because they are remote? If so, why was an example of a nearby monitor not included? Was your point to show that the smoke

was widespread? Pick more, not less sites. Was Colorado Springs impacted? It is not necessary to show all the sites, but just clarify your rational and pick sites to make your point and then say why you picked them.

43) Lines 414-416. What are you trying to say here? Can you refer to Figure 4? Are you trying to say the smoke was widespread? If so, say that and present evidence.

44) Line 424-6. You should state this much more strongly and earlier on. It is a major conclusion of the paper. You have direct evidence of this variability.

45) Figure S9-S10. Are these figure needed? Could they be combined with Figure 4 and used together as supporting evidence?

46) Figure S11. Is this figure needed? Could you just state the values in the text? An entire figure for two data points with error bars is excessive.

47) References. Please include reference showing where the public data you used came from (CDPHE, Forest Service, NPS).

Minor Issues/Typos

1) Line 62. The use of the pronoun "they" is vague. Please clarify the wording.

2) Line 76. The term "This region" is vague and should be made more specific and the wording should be clarified.

3) Line 220. There should be a comma after "however"

4) Line 243. A comma is needed after "Thus"

5) Line 278. A comma is needed after "Thus"

6) Line 286. This is important and should be emphasized more if possible, rather than burying it deep in a paragraph. Perhaps making the PAN and alkyl nitrate discussions separate paragraphs would clarify enough.

7) Line 300. Phrasing is a bit confusing. Rather than saying "fewer days" which is a

little vague here, rephrase saying that period 1 had a shorter duration than period 2, or the equivalent.

8) Line 302. "more significant changes"...than what? NO? Clarify the wording here.

9) Line 310. The phrase "is one hypothesis" is awkward. I suggest rephrasing this sentence.

10) Line 317. A comma is needed after "In this section"

11) Line 419. "Very high" is not specific enough. Include the value here.

12) P 6, line 161-166. Section 3: Things like red triangles, black lines, etc. should be in the figure caption, but not paper text. Only science/discussion should be in the paper body. Also, what are the blue circles? Not in legend or caption. There should be a space between 1000 and m on line 166.

---

## Referee Comment (RC2) · Anonymous Referee #2 · 27 Apr 2017

Lindaas et al. measured the influence of transported biomass burning smoke on atmospheric composition in the Colorado Front Range. Their study included measurements for an impressive range of compounds, including VOCs, reactive nitrogen, and ozone; the field work seems to have been carefully performed. The authors assessed several meteorological variables and determined that they were not the cause of the changes observed during the smoke-impacted periods.

Unfortunately, however, the manuscript largely reads like a list of observations without clear conclusions, particularly sections 4.1 and 4.2 (a few specific examples are noted below). The authors generally devote a large chunk of the text trying to rule out explanations other than biomass burning for a given observation (which is fine), but they

never seem to circle back and discuss clearly how their results contribute new insights into the "impact of aged wildfire smoke on atmospheric composition". What is the significance of the observed changes beyond that they can be attributed to smoke and not meteorology?

Additionally, each species (or class of compounds) is generally discussed independently of the others, with minimum consideration of the overall chemical system. For example, from the Introduction, I expected the measurements of the ozone precursors to inform the observed changes in ozone during the smoke influenced periods, yet section 4.3 focuses solely on the ozone data except for one brief mention of VOCs on line 396. How do the observations all link together?

Further, it is not clear why valid data is omitted from the discussion for the July smoke-influence period (i.e., CO, CH4). Also, why was only a small subset of the 40+ measured VOCs included in the manuscript, especially when many of the compounds in that subset have high emissions from other sources in the region and displayed no average change between conditions with and without smoke influence? The authors should more clearly justify their decisions when focusing on only a fraction of the available data (and ideally include the extra data in the supplement for evaluation).

Lastly, comparison of the observations presented in this manuscript to previous studies of transported/aged biomass burning is needed. There are many more relevant publications than the authors seem to give credit (lines 67-68). A few examples: (Jaffe et al., 2004; Mauzerall et al., 1998; Wotawa and Trainer, 2000; de Gouw et al., 2004) and additional works cited in (Heilman et al., 2014).

For these reasons, I think the paper is in need of substantial revisions before I can recommend publication in ACP.

Specific comments:

Line 60: State the EPA ozone standard.

Lines 102-106: Basic details of the GC system are missing. Was it a GC-MS? GC-FID? How were air samples trapped and introduced onto the column(s)? Over what time period? Which compounds were included in the calibration mixture? What is the uncertainty associated with the measurement?

Lines 159-163: By what metric and threshold does the HMS smoke product determine smoke impact? More explanation is needed here given that the field sites are just outside of the grey shaded "smoke-influenced" regions on July 7 (Figure 2), suggesting less relative smoke impact than the August time period where BAO is in the middle of smoke-impacted region; and yet the concentrations of CO and PM2.5 are significantly higher during the July period compared to the August period (Figure 1). Are data from any additional air monitoring sites available along those air mass trajectories to better establish that the air was indeed originating from regions more strongly influenced by smoke during the July period?

Lines 177-178: "we did not quantify species with known large
biomass burning emission ratios (e.g. hydrogen cyanide, acetonitrile, most oxygenated organic species)". Were these compounds not quantified or not measured/detected?
If acetonitrile and HCN were detected (even if not directly calibrated), then it is puzzling that they were omitted from the discussion, as these compounds are more specific biomass burning tracers than CO and PM2.5, with lifetimes much longer than the transport time of the air masses. Could their GC peak areas at least be used to determine relative differences between the periods with and without smoke influence? If no significant changes were observed in the peak areas for these markers, then the implications of that for assessing biomass burning influence need to be discussed. If HCN and acetonitrile were not or could not be observed under the GC operating conditions, please clarify the text. The quoted sentence also needs a citation.

Line 180-181: It's clear why the July period was omitted for the VOCs, but why were the CO and CH4 measurements for the July time period also omitted from the discussion? Those species were not subject to the GC issues. From Figure 1, CO had a much larger

enhancement during July vs. August smoke-impacted periods... were the differences between the two periods driven by meteorology, fire size, or other factors? Did methane behave similarly? This seems to be a missed opportunity for an interesting comparison.

Lines 198-202: "Average enhancements of CH4 were a much smaller percentage of (∼3% or 67 ppbv), but comparable in magnitude to, the
CO mixing ratio enhancement." Rephrase this sentence so the meaning is clear... I believe the percentage is meant to give the CH4 enhancement during the smoke impacted periods, but that is not how the sentence reads. Is the observed CH4 enhancement of 3% statistically significant given that the stated uncertainty in the CH4 measurement is 6% (line 99)? Also, the emission factor of CO is generally >10x that of CH4 from biomass burning (Akagi et al., 2011), thus it is curious that the CH4 enhancement is "comparable in magnitude to the CO enhancement" and could suggest that the other local sources are dominant. Overall, it is not clear from the discussion if the authors attribute the observed 67ppb CH4 enhancement to biomass burning influence or what conclusions should be drawn from the methane observations.

Lines 215-224: Why is the dampening of the diurnal cycle amplitudes for the alkanes significant if there was no net enhancement of these compounds during the smoke-influenced periods (line 210)? Were the changes at certain times of day due to biomass burning influence? What is a possible explanation if not changes in PBL height? The take home message of this discussion regarding the impacts of aged wildfire smoke on the diurnal cycles isn't clear. (Similar questions apply to the discussion of diurnal cycles for the other compounds, as well.) Also, please include the ethane diurnal cycles in Figure 5 or the supplement if they warrant this much discussion. It is difficult to follow the text without seeing the relevant diurnal cycle data.

Lines 236-237: Needs a citation. Also, have the authors considered that decreased alkene abundances locally were due to the increased ozone rather than OH? The abundance of aromatics with similar OH reactivity to ethene (Atkinson and Arey, 2003), but negligible O3 reactivity (toluene, xylene, ethyl benzene) did not change during the

smoke-impacted period (lines 257-259). If increased OH oxidation was the cause of the decreased alkene abundances, then shouldn't the aromatics have been similarly influenced? Perhaps a broader discussion of the relative sources and their strengths for the various compounds would also help the discussion.

Lines 283-287: What is the significance of the PPN/PAN ratio?

Lines 302-304: The $NO_2$ diurnal cycles during the July smoke period and the smoke-free period shown in Figure 5c are nearly identical. Are the differences discussed here statistically significant and/or important?

Lines 308-309: It has not been explained anywhere that PAN is a reservoir for $NO_x$. Some readers may be confused.

Lines 368-370: "we found the same enhancement in O3 for a
given temperature when comparing smoke-impacted observations to smoke-free observations assigned to this cluster as we
found for the complete dataset (Figures S9 and S10)." First, how can there be fewer datapoints within the "complete dataset" (N=30, Figure S10) than a cluster (N=33, Figure S9a)? Or should the complete dataset instead refer to Figure 6? In which case, the data do not support the claim. There is no discernible difference between smoke-free and smoke-impacted cases in Figure S9, certainly not a 10ppb increase on average for the smoke-influenced periods. Second, why was this cluster analysis limited to just 12:00-17:00? The northwesterly flow cluster was the only one with a meaningful number of data points during the smokey period, so why not use all of the data for a more robust comparison across the trajectory clusters?

Line 373: Include a citation.

Line 374-377: Is 65 ppbv MDA8 a formal definition of "high" ozone or was it defined by the authors? If the latter, why was this value chosen as a benchmark over the NAAQS value of 70ppb? Also, add more context for how these observations relate to the broader trends in the Colorado Front Range. How many ozone exceedance days

are typical in the in this region annually? Is the frequency of high ozone days shown in Figure 7 a departure from "normal" conditions?

Line 395: Include a citation and brief description for OPE.

Line 397-398: "Fully addressing the question of whether the smoke enhanced local O3 production in the polluted Front Range
requires the use of a chemical transport model, and is beyond the scope of this work."
There could still be some attempt made to qualitatively link together the observations for the precursors and resulting changes in ozone, which would go a long way toward improving the manuscript. In general, more consideration of chemistry in addition to meteorological variables would help.

Figure 3. Out of the 40+ VOCs measured, why were these compounds chosen when most of them have other large sources in the area? Instead of the binary color scheme, can a colorscale be applied to show the percent change for each species?

Figure 5: Do the color bands represent one standard deviation of each average diurnal cycle?

Technical corrections:

Line 94 (and elsewhere): "1 $\mu$m PTFE filter membrane" Do you mean the pore size, not the filter size, was 1 um?

Figure 1: Please include more tick marks on the date axis so that specific dates can be located on the traces.

Figure 6: Can Fig. S4 be merged with this one so all of the data is included in a single plot?

Figures S8 and S9: Arrange the panels in the same order.

Figures S9 and S10: Note more clearly in the caption which data are shown (e.g., afternoon only?). Also include labels for the data in the legend, not just the number of

points.

References:

Akagi, S. K., Yokelson, R. J., Wiedinmyer, C., Alvarado, M. J., Reid, J. S., Karl, T., Crounse, J. D., and Wennberg, P. O.: Emission factors for open and domestic biomass burning for use in atmospheric models, Atmos. Chem. Phys., 11, 4039-4072, DOI 10.5194/acp-11-4039-2011, 2011.

Atkinson, R., and Arey, J.: Atmospheric Degradation of Volatile Organic Compounds, Chemical Reviews, 103, 4605-4638, 2003.

de Gouw, J. A., Cooper, O. R., Warneke, C., Hudson, P. K., Fehsenfeld, F. C., Holloway, J. S., Hubler, G., Nicks, D. K., Nowak, J. B., Parrish, D. D., Ryerson, T. B., Atlas, E. L., Donnelly, S. G., Schauffler, S. M., Stroud, V., Johnson, K., Carmichael, G. R., and Streets, D. G.: Chemical composition of air masses transported from Asia to the U. S. West Coast during ITCT 2K2: Fossil fuel combustion versus biomass-burning signatures, J. Geophys. Res. Atmos., 109, Artn D23s20 10.1029/2003jd004202, 2004.

Heilman, W. E., Liu, Y. Q., Urbanski, S., Kovalev, V., and Mickler, R.: Wildland fire emissions, carbon, and climate: Plume transport, and chemistry processes, Forest Ecol Manag, 317, 70-79, 10.1016/j.foreco.2013.02.001, 2014.

Jaffe, D., Bertschi, I., Jaegle, L., Novelli, P., Reid, J. S., Tanimoto, H., Vingarzan, R., and Westphal, D. L.: Long-range transport of Siberian biomass burning emissions and impact on surface ozone in western North America, Geophys. Res. Lett., 31, Artn L16106 Doi 10.1029/2004gl020093, 2004.

Mauzerall, D. L., Logan, J. A., Jacob, D. J., Anderson, B. E., Blake, D. R., Bradshaw, J. D., Heikes, B., Sachse, G. W., Singh, H., and Talbot, B.: Photochemistry in biomass burning plumes and implications for tropospheric ozone over the tropical South Atlantic, J. Geophys. Res. Atmos., 103, 8401-8423, 10.1029/97jd02612, 1998.

Wotawa, G., and Trainer, M.: The influence of Canadian forest fires on pollutant concentrations in the United States, Science, 288, 324-328, 10.1126/science.288.5464.324, 2000.

---

## Author Comment (AC2) · 30 Jun 2017

*Thank you to this reviewer for their time and detailed comments. We feel that addressing each of these comments has led to a more precise and improved manuscript. Below our responses and excerpted text to reviewer comments are in italics.*

Lindaas et al. measured the influence of transported biomass burning smoke on atmospheric composition in the Colorado Front Range. Their study included measurements for an impressive range of compounds, including VOCs, reactive nitrogen, and ozone; the field work seems to have been carefully performed. The authors assessed several meteorological variables and determined that they were not the cause of the changes observed during the smoke-impacted periods. Unfortunately, however, the manuscript largely reads like a list of observations without clear conclusions, particularly sections 4.1 and 4.2 (a few specific examples are noted below). The authors generally devote a large chunk of the text trying to rule out explanations other than biomass burning for a given observation (which is fine), but they never seem to circle back and discuss clearly how their results contribute new insights into the "impact of aged wildfire smoke on atmospheric composition". What is the significance of the observed changes beyond that they can be attributed to smoke and not meteorology?

*We thank the reviewer for their thoughts on how to better focus the paper. We had assumed that most readers would immediately ask if meteorological anomalies could be responsible for the changes observed. However, it seems like we may have provided more information than necessary on this topic. We have re-structured the conclusions to better summarize our findings, and we have removed much of the back-up meteorological analysis that supports our conclusion that some of the unique findings must be due to the presence of smoke.*

*The strength of this paper is that it shows two examples of how a subset of ozone precursors changes in the presence of aged fire smoke. The dataset is interesting because of the high quality of the observations, but it is also interesting because the fires responsible for the smoke in August 2015 were extreme. The 2015 Washington wildfires season was the largest in history. This paper demonstrates that ozone during both the July and August smoke-impacted periods was higher than expected based on ambient temperatures (i.e. for a given temperature average hourly ozone is greater during the smoke-impacted periods than the smoke-free period. The paper also shows which ozone precursors also change in the presence of smoke. We do not understand the mechanisms driving all these changes.*

Additionally, each species (or class of compounds) is generally discussed independently of the others, with minimum consideration of the overall chemical system. For example, from the Introduction, I expected the measurements of the ozone precursors to inform the observed changes in ozone during the smoke influenced periods, yet section 4.3 focuses solely on the ozone data except for one brief mention of VOCs on line 396. How do the observations all link together?

*We agree that it would be ideal to tie this together better, but that would require additional observations in addition to the use of a chemical transport model that represents smoke processes well (which many models struggle with currently). We don't believe that we have the ideal suite of constraints in our measurements. For example we are missing observations of nighttime radical sources and $J_{NO2}$, both of which would be useful in testing different hypothesized mechanisms for the larger $NO_2$ during the morning and evening smoke-impacted periods. Additionally, we only observed a limited suite of oxygenated species. We also have no constraints on the gas-phase emissions of this particular fire complex with which to constrain the evolution of the plume. We feel that providing a specific chemical mechanism for the ozone production within the plume during its transport to BAO would be speculative at best.*

Further, it is not clear why valid data is omitted from the discussion for the July smoke influence period (i.e., CO, CH4). Also, why was only a small subset of the 40+ measured VOCs included in the manuscript, especially when many of the compounds in that subset have high emissions from other sources in the region and displayed no average change between conditions with and without smoke influence? The

authors should more clearly justify their decisions when focusing on only a fraction of the available data (and ideally include the extra data in the supplement for evaluation).

*We actually were as inclusive as possible here. No available data from the field intensive was omitted. We have added a table to the SI that shows the abundance of the VOCs. The only significant changes that we observed in the VOCs were those included in Figure 3. We are not able to probe changes in composition as extensively for the smoke-impacted period in July because that technically occurred before the start of our field campaign. We were fortunate that many of the "easy" measurements (i.e. ozone) were running already at that time, but the more labor-intensive instruments (i.e. the gas chromatographs used for the VOC measurements) were not running. The dataset is interesting because of the high quality of the observations, but it is also interesting because the fires responsible for the smoke in August 2015 were extreme. The 2015 Washington wildfires season was the largest in history. There are a number of case studies, with high chemical specificity, of aged wildfires smoke. However, there are very few measurements of this duration (i.e. aircraft will sample a plume over the course of a few hours) or within a polluted boundary layer.*

Lastly, comparison of the observations presented in this manuscript to previous studies of transported/aged biomass burning is needed. There are many more relevant publications than the authors seem to give credit (lines 67-68). A few examples: (Jaffe et al., 2004; Mauzerall et al., 1998; Wotawa and Trainer, 2000; de Gouw et al., 2004) and additional works cited in (Heilman et al., 2014).

*We thank the reviewer for noting these papers, and we have added references. All the earlier papers are cited in the Jaffe and Wigder, 2012, review paper that we cite. A key difference here is that these plumes were largely sampled in the free troposphere, and not mixed with polluted boundary layer air. Our study is very unique in the length of time that the smoke was sampled (nearly 14 days). This is a very large number of samples of an aged plume over a long time period. This type of extensive sampling is not possible from an aircraft.*

*"There are well-documented case studies of within plume $O_3$ production (see Jaffe and Wigder (2012); Heilman et al. (2014), and references within) and time periods where smoke contributed to exceedances of the U.S. EPA National Ambient Air Quality Standard (NAAQS) for $O_3$ (Morris et al., 2006; Pfister et al., 2008), currently a maximum daily 8 hour average of 70 ppbv."*

For these reasons, I think the paper is in need of substantial revisions before I can recommend publication in ACP.

*Thank you for your thorough reading of the manuscript, we feel that we have been able to address all the comments below.*

Specific comments:
Line 60: State the EPA ozone standard.

*This line has been edited. See below for the new text.*

*"…time periods where smoke contributed to exceedances of the U.S. EPA National Ambient Air Quality Standard (NAAQS) for $O_3$ (Morris et al., 2006; Pfister et al., 2008), currently a maximum daily 8 hour average of 70 ppbv."*

Lines 102-106: Basic details of the GC system are missing. Was it a GC-MS? GCFID? How were air samples trapped and introduced onto the column(s)? Over what time period? Which compounds were included in the calibration mixture? What is the uncertainty associated with the measurement?

*We have provided answers to the reviewers questions as edits to the text and continue to point to the full description of the GC instrument in Abeleira et al. (2017). The revised text is below.*

*"A custom 4-channel cryogen-free gas chromatography (GC) system (Sive et al., 2005) was used to measure selected non-methane hydrocarbons (NMHCs), $C_1 – C_2$ halocarbons, alkyl nitrates (ANs), and oxygenated volatile organic compounds (OVOCs) at sub-hourly time resolution; approximately one sample every 45 minutes. The inlet was located at 6 m a.g.l. with a 1 μm pore size teflon filter. Ambient air for each sample was collected and preconcentrated over 5 minutes, with a one liter total sample volume. A calibrated whole air mixture was sampled in the field after every ten ambient samples to monitor sensitivity changes and measurement precision. A full description of this instrument and the associated uncertainties for each detected species is provided in (Abeleira et al., 2017)."*

Lines 159-163: By what metric and threshold does the HMS smoke product determine smoke impact? More explanation is needed here given that the field sites are just outside of the grey shaded "smoke-influenced" regions on July 7 (Figure 2), suggesting less relative smoke impact than the August time period where BAO is in the middle of smoke-impacted region; and yet the concentrations of CO and PM2.5 are significantly higher during the July period compared to the August period (Figure 1). Are data from any additional air monitoring sites available along those air mass trajectories to better establish that the air was indeed originating from regions more strongly influenced by smoke during the July period?

*As discussed in response to the other reviewer, the HMS smoke product uses data from multiple NOAA and NASA satellites to identify smoke-plumes in the atmospheric column. The smoke is detected using visible imagery assisted by infrared imagery, which allows clouds and smoke to be distinguished. The HMS smoke product is a conservative estimate of the smoke because for smoke to be identified, it has to be visible from satellite. A comprehensive description of the HMS smoke product is available in Brey et al. [2017], currently under review in ACPD. There are also additional earlier references within Brey et al. (2017) that also describe this operational product. We have added this information to the text, see below.*

*"The NOAA Hazard Mapping System smoke polygons (grey shading) show that the smoke events observed at BAO were large regional events. The HMS smoke product is produced using multiple NASA and NOAA satellite products (Rolph et al., 2009). Smoke in the atmospheric column is detected using both visible and infrared imagery and is fully described in Brey et al. (2017). The extent of smoke plumes within the HMS dataset represents a conservative estimate, and no information is provided on the vertical extent or vertical placement of the plumes."*

*Brey, S. J., Ruminski, M., Atwood, S. A., and Fischer, E. V.: Connecting smoke plumes to sources using Hazard Mapping System (HMS) smoke and fire location data over North America, Atmos. Chem. Phys. Discuss., https://doi.org/10.5194/acp-2017-245, in review, 2017.*

*There were actually plenty of additional air monitoring sites available along the trajectory of the smoke to establish that the air was indeed originating from regions more strongly influenced by smoke. For example, here is a map showing the location of the fires identified by HMS analysts. We have also plotted the overlapping smoke plumes for that day. HMS does provide contours of concentration, but they are approximate. The colored dots show the locations of PM2.5 monitors throughout the western U.S. You can see that PM was moderate to unhealthy within the plume. When viewing this figure, please keep in mind that the HMS smoke plumes show smoke in the column, not necessarily at the surface. The concentration of PM at the surface will depend on how much of the smoke mixes into the boundary layer. This makes it easy to explore data associated with this even for any region of choice. We have not developed a larger paper on these fires, specifically addressing impacts on composition upwind of Colorado, because we are aware of other groups doing these types of more broad analyses. We decided to focus on our unique set of observations. The figure below was produced using a web-application that we have developed. https://stevenjoelbrey.shinyapps.io/HMSExplorer/*

[Figure]

*Here is the comparable figure for the August event. You can see that surface PM enhancements were much higher closer to the source fires in this case. You can also see that there were fires in Washington and Idaho, similar to what we already show with the MODIS hotspots in Figure 2. You can see that surface PM was enhanced across the intermountain west during this time.*

[Figure]

Lines 177-178: "we did not quantify species with known large biomass burning emission ratios (e.g. hydrogen cyanide, acetonitrile, most oxygenated organic species)". Were these compounds not quantified or not measured/detected? If acetonitrile and HCN were detected (even if not directly calibrated), then it is puzzling that they were omitted from the discussion, as these compounds are more specific biomass burning tracers than CO and PM2.5, with lifetimes much longer than the transport time of the air masses. Could their GC peak areas at least be used to determine relative differences between the periods with and without smoke influence? If no significant changes were observed in the peak areas for these markers, then the implications of that for assessing biomass burning influence need to be discussed. If HCN and acetonitrile were not or could not be observed under the GC operating conditions, please clarify the text. The quoted sentence also needs a citation.

*Our GC system was not set up to detect HCN and acetonitrile. Since we did not anticipate sampling wildfire smoke and the focus of the campaign was to assess anthropogenic ozone precursors in the Colorado Front Range, the GC was optimized to be sensitive to the light alkanes, alkenes, and a few OVOCs along with a handful of alkyl nitrates. The chromatograms were checked for HCN and acetonitrile peaks after the campaign but those peaks were not able to be identified.*

*We have edited these lines to be more specific, see below.*

*"The focus of the BAO field intensive was to study the photochemistry of local emissions from oil and gas development (e.g. Gilman et al., 2013; Swarthout et al., 2013; Thompson et al., 2014; Abeleira et al., 2017), and the GC system was not set up to quantify species with known large biomass burning emission ratios (e.g. hydrogen cyanide, acetonitrile, most oxygenated organic species) (Akagi et al., 2011). The chromatograms were checked for HCN and acetonitrile peaks after the campaign but those peaks were not able to be identified."*

Line 180-181: It's clear why the July period was omitted for the VOCs, but why were the CO and CH4 measurements for the July time period also omitted from the discussion? Those species were not subject to the GC issues. From Figure 1, CO had a much larger enhancement during July vs. August smoke-impacted periods... were the differences between the two periods driven by meteorology, fire size, or other factors? Did methane behave similarly? This seems to be a missed opportunity for an interesting comparison.

*The authors appreciate the reviewer catching this oversight. CO was shown in Figure 1, but not specifically discussed. We have added the quantified changes in CO and CH4 during the July smoke period to the discussion, and mention one possible reason for the observed differences in CO and PM between the two smoke periods. The edited text is shown below.*

*"Mean hourly CO mixing ratios were significantly enhanced by 223 ppbv, or 170% during the July smoke-impacted period and by 92 ppbv, or 70%, during the August smoke-impacted period (Figure 1). This enhancement was present across the diurnal cycle (Figure 3) and a both smoke periods displayed a higher range of CO mixing ratios (July: 127 – 639 ppbv, August: 101 – 529 ppbv, smoke-free: 72 – 578 ppbv). The two smoke periods differed in their sources fires, length, and meteorology, with higher average CO and $PM_{2.5}$ measurements in the July smoke period (Figure 1)."*

Lines 198-202: "Average enhancements of CH4 were a much smaller percentage of (∼3% or 67 ppbv), but comparable in magnitude to, the CO mixing ratio enhancement." Rephrase this sentence so the meaning is clear. . . I believe the percentage is meant to give the CH4 enhancement during the smoke impacted periods, but that is not how the sentence reads. Is the observed CH4 enhancement of 3% statistically significant given that the stated uncertainty in the CH4 measurement is 6% (line 99)? Also, the emission factor of CO is generally >10x that of CH4 from biomass burning (Akagi et al., 2011), thus it is curious that the CH4 enhancement is "comparable in magnitude to the CO enhancement" and could suggest that the other local sources are dominant. Overall, it is not clear from the discussion if the authors attribute the observed 67ppb CH4 enhancement to biomass burning influence or what conclusions should be drawn from the methane observations.

*We agree with the reviewer that this was confusing as originally written. We have re-written this section to read:*

*"Average enhancements of $CH_4$ were similar for both periods (July: 52 ppbv, August: 50 ppbv, or ~ 2.5% increase). Methane has a relatively high background at BAO due to large emissions of $CH_4$ in nearby Weld County from livestock production and oil and gas development (Pétron et al., 2014; Townsend-Small et al., 2016). Taken together, the larger background of $CH_4$ and the large local sources of $CH_4$ in the Front Range served to mute the impact of the August smoke on overall $CH_4$ abundances. The diurnal cycle of $CH_4$ did not change during the smoke-impacted period as compared to the smoke-free period and we observed a similar range of mixing ratios (~1,840 – 3,360 ppbv) in the both smoke-free and smoke-impacted periods.*

*We note several large spikes in CH₄ on the order of minutes during the August smoke-impacted period, but we do not believe that these are related to the presence of smoke because they were not correlated with similar excursions in CO and PANs, and exhibited strong correlations with propane and other tracers of oil and gas and other anthropogenic activity."*

Lines 215-224: Why is the dampening of the diurnal cycle amplitudes for the alkanes significant if there was no net enhancement of these compounds during the smoke influenced periods (line 210)? Were the changes at certain times of day due to biomass burning influence? What is a possible explanation if not changes in PBL height? The take home message of this discussion regarding the impacts of aged wildfire smoke on the diurnal cycles isn't clear. (Similar questions apply to the discussion of diurnal cycles for the other compounds, as well.) Also, please include the ethane diurnal cycles in Figure 5 or the supplement if they warrant this much discussion. It is difficult to follow the text without seeing the relevant diurnal cycle data.

*The discussion on diurnal cycles was meant to be part of the documentation of any and all changes we observed. The authors agree with the reviewer that there is not a clear take home message about the alkane diurnal cycles at this point. Thus, for clarity, this section and associated discussion has been removed for the revised paper.*

Lines 236-237: Needs a citation. Also, have the authors considered that decreased alkene abundances locally were due to the increased ozone rather than OH? The abundance of aromatics with similar OH reactivity to ethene (Atkinson and Arey, 2003), but negligible O3 reactivity (toluene, xylene, ethyl benzene) did not change during the smoke-impacted period (lines 257-259). If increased OH oxidation was the cause of the decreased alkene abundances, then shouldn't the aromatics have been similarly influenced? Perhaps a broader discussion of the relative sources and their strengths for the various compounds would also help the discussion.

*Thanks to the reviewer for an additional hypothesis that we not consider earlier. The discussion of hypotheses for the decreased alkene abundances has been expanded. See edited section below.*

*"The atmospheric lifetimes of the four alkenes we quantified (isoprene, propene, ethene, and cis-2-butene) range from tens of minutes to hours. Surprisingly, we observed significant decreases in the abundance of isoprene, propene and ethene during the August smoke-impacted period compared to the smoke-free period: -64% (-143 pptv), -77% (-39 pptv), and -81% (-206 pptv) respectively (for summary statistics see Table 1). The shape of the diurnal cycles did not change (Figure S1), though propene and ethene were near their respective limits of detection for the majority of each day during the smoke-impacted period. Given the short lifetimes of these species, this indicates that the presence of the smoke changed either local anthropogenic or biogenic emissions of these species, or their respective rates of oxidation by OH or O₃. We present several potential mechanisms here, but we do not have sufficient information to determine if one of these is solely responsible for the pattern we observed.*

*Our first hypothesis is that fewer anthropogenic emissions of these alkenes drove the observed decreases in alkene abundances. However, there is no evidence that anthropogenic emissions were different during the August smoke-impacted period. Specifically, the August smoke-impacted period encompassed both weekdays and weekends and did not contain any state or federal holidays. Therefore we move to our second hypothesis, that changes in the biogenic emissions of alkenes accounted for the decreased alkene mixing ratios. Isoprene is widely known to be emitted by broad leaf vegetation, and emission rates are positively correlated with light and temperature (Guenther et al., 2006). Recent measurements quantified ethene and propene emissions from a ponderosa pine forest near Colorado Springs, CO, with an inter-daily light and temperature dependence similar to isoprene (Rhew et al., 2017). Interestingly, emissions and mixing ratios of ethene and propene were not closely correlated with isoprene within the diurnal cycle, indicating they have different vegetative/soil sources than isoprene at that site. Ponderosa pine stands are present in the foothills on the western edge of the plains in the Front Range, and several species of broad leaf trees are present along waterways, in urban areas, and in the foothills of this region. Thus, biogenic sources of ethene, propene, and isoprene in the region around BAO are reasonable. Given the August smoke-impacted period was on average colder than the smoke-free period, and potentially saw a reduction*

*in photosynthetic active radiation (PAR) at the surface due to the increased number of aerosols, it is possible that biogenic emissions of isoprene, ethane, and propene were suppressed. However, biogenic fluxes of these compounds are unavailable for the region around BAO during summer 2015, and extrapolating emissions from one ponderosa pine stand to the rest of the Front Range may be overly ambitious. Further, we note that a PMF analysis of the VOC data from this site did produce a 'biogenic factor' dominated by isoprene, but with negligible contribution of any other hydrocarbon, suggesting that the biogenic component of these $C_2$-$C_3$ alkenes was small (Abeleira et al…). Thus, while the hypothesis that smoke suppressed biogenic emissions remains feasible, we will consider other potential causes for the observed decrease in alkene abundances.*

*The alkenes we measured all have high reactivities with respect to OH ($> 8 \times 10^{12}$ molec$^{-1}$ cm$^3$ s) and $O_3$ ($> 0.1 \times 10^{17}$ molec$^{-1}$ cm$^3$ s) (Atkinson and Arey, 2003). Enhancements in OH abundances have been inferred in wildfire smoke plumes by several studies (e.g. Akagi et al. (2012); Hobbs et al. (2003); Liu et al. (2016); Yokelson et al. (2009)). If the August smoke-impacted period was characterized by higher than normal OH mixing ratios, then a third hypothesis is that the observed decreases in alkene abundances could be due to a higher oxidation rate by OH due to higher OH concentrations. However, other measured VOCs such as o-xylene or methylcyclohexane have similar OH reactivities to ethene (Atkinson and Arey, 2003), and we do not see associated decreases in abundances of these other VOCs. Thus, the hypothesis of increased oxidation by OH causing decreased alkene abundances in the August smoke period is not supported by the full suite of measurements at BAO.*

*Lastly, we move on to our final hypothesis. Alkenes have much higher rates of reaction with $O_3$ than the other VOCs we quantified. As we will demonstrate in Section 4.3, the August smoke-impacted period was characterized by higher $O_3$ abundances than would otherwise be expected. Therefore, the fourth hypothesis regarding decreased alkene abundances is that enhanced alkene oxidation by $O_3$ decreased the observed mixing ratios. Two factors complicate this hypothesis though. First, we do not observe a negative relationship between $O_3$ and alkene abundance during the smoke-free time periods (i.e. increased $O_3$ is not correlated with decreased alkenes when no smoke is present). Second, despite having a higher reaction rate with $O_3$ compared to propene and ethene, cis-2-butene does not decrease during the August smoke-impacted period.*

*After careful consideration, there is no strong evidence supporting any of these four hypotheses over the others (suppressed anthropogenic emissions, suppressed biogenic emissions, increased OH, increased $O_3$). It is possible that more than one of these processes could have contributed to the observation of decreased alkene abundances during the 2 week-long August smoke-influenced period. Future field campaigns and modeling work are necessary to understand how common suppressed alkene abundances may be in smoke-impacted airmasses, and what processes might control this phenomenon. "*

*Akagi, S. K., Craven, J. S., Taylor, J. W., McMeeking, G. R., Yokelson, R. J., Burling, I. R., Urbanski, S. P., Wold, C. E., Seinfeld, J. H., Coe, H., Alvarado, M. J., and Weise, D. R.: Evolution of trace gases and particles emitted by a chaparral fire in California, Atmos. Chem. Phys., 12, 1397-1421, 10.5194/acp-12-1397-2012, 2012.*

*Hobbs, P. V., Sinha, P., Yokelson, R. J., Christian, T. J., Blake, D. R., Gao, S., Kirchstetter, T. W., Novakov, T., and Pilewskie, P.: Evolution of gases and particles from a savanna fire in South Africa, Journal of Geophysical Research: Atmospheres, 108, n/a-n/a, 10.1029/2002JD002352, 2003.*

*Liu, X., Zhang, Y., Huey, L. G., Yokelson, R. J., Wang, Y., Jimenez, J. L., Campuzano-Jost, P., Beyersdorf, A. J., Blake, D. R., Choi, Y., St. Clair, J. M., Crounse, J. D., Day, D. A., Diskin, G. S., Fried, A., Hall, S. R., Hanisco, T. F., King, L. E., Meinardi, S., Mikoviny, T., Palm, B. B., Peischl, J., Perring, A. E., Pollack, I. B., Ryerson, T. B., Sachse, G., Schwarz, J. P., Simpson, I. J., Tanner, D. J., Thornhill, K. L., Ullmann, K., Weber, R. J., Wennberg, P. O., Wisthaler, A., Wolfe, G. M., and Ziemba, L. D.: Agricultural fires in the southeastern U.S. during SEAC4RS: Emissions of trace gases and particles and evolution of ozone, reactive nitrogen, and organic aerosol, Journal of Geophysical Research: Atmospheres, n/a-n/a, 10.1002/2016JD025040, 2016.*

*Yokelson, R. J., Crounse, J. D., DeCarlo, P. F., Karl, T., Urbanski, S., Atlas, E., Campos, T., Shinozuka, Y., Kapustin, V., Clarke, A. D., Weinheimer, A., Knapp, D. J., Montzka, D. D., Holloway, J., Weibring, P., Flocke, F., Zheng, W., Toohey, D., Wennberg, P. O., Wiedinmyer, C., Mauldin, L., Fried, A., Richter, D., Walega, J., Jimenez, J. L., Adachi, K., Buseck, P. R., Hall, S. R., and Shetter, R.: Emissions from biomass burning in the Yucatan, Atmos. Chem. Phys., 9, 5785-5812, 10.5194/acp-9-5785-2009, 2009.*

Lines 283-287: What is the significance of the PPN/PAN ratio?

*In response to Reviewer 1's comments, we have removed these sentences.*

Lines 302-304: The NO2 diurnal cycles during the July smoke period and the smoke free period shown in Figure 5c are nearly identical. Are the differences discussed here statistically significant and/or important?

*The authors included the discussions of NO2 diurnal cycles during the July smoke period in the spirit of documenting any statistically significant changes in the dataset between smoke-impacted and smoke-free periods. However, since there are no obviously testable hypotheses for the observed changes, the authors have chosen to omit this discussion in the revised paper. The revised section is below.*

*"During the July smoke-impacted period, $NO_2$ was within the range of smoke-free measurements. In contrast $NO_2$ during the August smoke-impacted period followed the same diurnal cycle but had pronounced significant increases in average mixing ratios during the morning and evening hours of ~8 ppbv (17%) following sunrise and 3 ppbv (60%) following sunset. "*

Lines 308-309: It has not been explained anywhere that PAN is a reservoir for NOx. Some readers may be confused.

*The authors thank the reviewer for pointing this out. This sentence has been edited to make this fact clear.*

*"Another hypothesis concerns the equilibrium between PAN and $NO_2$. The thermal decomposition of PAN can be a source of $NO_2$ (Singh and Hanst, 1981), but the concurrently observed PAN abundances during the August smoke-impacted period can only account for at most 1 ppbv of additional $NO_2$. PAN abundances were likely higher in the fresher plume, but still not likely sufficient to be the sole source of the additional $NO_2$."*

Lines 368-370: "we found the same enhancement in O3 for a given temperature when comparing smoke-impacted observations to smoke-free observations assigned to this cluster as we found for the complete dataset (Figures S9 and S10)." First, how can there be fewer datapoints within the "complete dataset" (N=30, Figure S10) than a cluster (N=33, Figure S9a)? Or should the complete dataset instead refer to Figure 6? In which case, the data do not support the claim. There is no discernible difference between smoke-free and smoke-impacted cases in Figure S9, certainly not a 10ppb increase on average for the smoke-influenced periods. Second, why was this cluster analysis limited to just 12:00-17:00? The northwesterly flow cluster was the only one with a meaningful number of data points during the smokey period, so why not use all of the data for a more robust comparison across the trajectory clusters?

*The comparison is meant to be between each cluster and the complete dataset in Figure 6. The authors agree with the reviewer that since Figure 6 makes use of all hours, Figures S9 and S10 should plot all hours as well. We have updated the Figures in the SI, and stand by our conclusions.*

Line 373: Include a citation.

*This section has been revised in light of the change in focus from MDA8 as the definition of high ozone to the 95[th] percentile of daytime hourly average ozone values. This change is discussed more thoroughly in the response to the next question in this review.*

Line 374-377: Is 65 ppbv MDA8 a formal definition of "high" ozone or was it defined by the authors? If the latter, why was this value chosen as a benchmark over the NAAQS value of 70ppb? Also, add more context for how these observations relate to the broader trends in the Colorado Front Range. How many ozone exceedance days are typical in the in this region annually? Is the frequency of high ozone days shown in Figure 7 a departure from "normal" conditions?

*In reviewing the decision to choose a definition for "high" ozone the authors have decided to follow the empirical definition outlined by Cooper et al., 2012, in their paper on ozone trends across the U.S. Cooper et al., 2012, define "high" ozone as an hourly average mixing ratio that is greater than the 95$^{th}$ percentile of all hourly average ozone mixing ratios during daytime (11am – 4pm local time) within a given study period. Applying this criteria to our dataset we define a "high ozone day" as any day in our dataset having at least one hourly average ozone mixing ratio above this 95$^{th}$ percentile value, calculated using all available data in our study period. This results in 9 days being defined as "high ozone days" within our study period, with 2 of them falling within the August smoke-impacted period. We have updated Figure 7 accordingly.*

*We feel this is the correct method for defining a high $O_3$ day for two reasons. First, BAO is not an EPA designated $O_3$ NAAQS site, and the BAO $O_3$ data are not explicitly calibrated to the EPA $O_3$ calibration scale. Thus, while we can calculate the MDA8 values for the BAO $O_3$ data, we do not feel comfortable comparing these values to sites designed for regulatory purposes. Second, our definition uses an empirical technique to define a high $O_3$ day, reducing the subjectivity associated with otherwise choosing a value and aligning our results more evenly with existing literature.*

*In terms of interannual context, for the months of July and August in each year 2009-2015 we calculated the number of days that had a maximum hourly average $O_3$ mixing ratio greater than the "high $O_3$ day" 95$^{th}$ percentile threshold (71.75 ppbv) in our study period. The average number of high $O_3$ days within those two months for a given year is 15.7. 2015 was lower than this, with 9 high $O_3$ days, and was the second lowest year after 2009.*

*The updated section with all this information is copied below.*

*"Following the definition in (Cooper et al., 2012), we define a "high $O_3$ day" as any day in our study period with at least one hour above the 95$^{th}$ percentile (71.75 ppbv) of all 11am – 4pm MDT hourly average $O_3$ measurements during the campaign. We found 9 individual high $O_3$ days during our study period, of which 2 occurred during the August smoke-impacted period (Figure 7). The total number of high $O_3$ days is lower than normal for the same time period in previous years. As we stated above, high $O_3$ during the August smoke period was not a result of abnormal meteorological variables, such as higher than normal temperatures. The lower portion of Figure 7 again shows that maximum daily temperatures during the smoke-impacted periods were the same as or lower than maximum daily temperatures during the smoke-free period."*

Line 395: Include a citation and brief description for OPE.

*We have updated the discussion of OPE to include the citation of Trainer et al., 1993, and to briefly define the term ozone production efficiency. See edited passage below.*

*"One measure of local production of $O_3$ is the ozone production efficiency (OPE). OPE is calculated as the slope of the relationship between $O_3$ and $NO_z$ (= $NO_y$ – $NO_x$) (Trainer et al., 1993). OPE is a measure of how the number of molecules of $O_3$ that are produced before a given $NO_x$ molecule is oxidized. To calculate OPE we used one minute $O_3$ and $NO_z$ data in 30 minute chunks from 12PM - 5PM MDT. The slopes were calculated using a reduced major axis regression (package lmodel2 for R software) and only OPE values corresponding to an $R^2 > 0.3$ were retained. We do not find any significant differences in average calculated OPE between the smoke-impacted (8 ± 3 ppbv/ppbv) and smoke-free periods (7 ± 3 ppbv/ppbv)."*

Line 397-398: "Fully addressing the question of whether the smoke enhanced local O3 production in the polluted Front Range requires the use of a chemical transport model, and is beyond the scope of this work." There could still be some attempt made to qualitatively link together the observations for the precursors and resulting changes in ozone, which would go a long way toward improving the manuscript. In general, more consideration of chemistry in addition to meteorological variables would help.

*We expanded our discussion of OPE and local ozone production in Section 4.3. See below for the added text.*

*"One measure of local production of $O_3$ is the ozone production efficiency (OPE). OPE is calculated as the slope of the relationship between $O_3$ and $NO_z$ $(= NO_y – NO_x)$ (Trainer et al., 1993). OPE is a measure of how the number of molecules of $O_3$ that are produced before a given $NO_x$ molecule is oxidized. To calculate OPE we used one minute $O_3$ and $NO_z$ data in 30 minute chunks from 12PM - 5PM MDT. The slopes were calculated using a reduced major axis regression (package lmodel2 for R software) and only OPE values corresponding to an $R^2 > 0.3$ were retained. We do not find any significant differences in average calculated OPE between the smoke-impacted $(8 \pm 3$ ppbv/ppbv) and smoke-free periods $(7 \pm 3$ ppbv/ppbv). Thus from the OPE perspective it does not appear there were any changes in the local production efficiency of $O_3$ due to the presence of smoke. On the other hand, we documented many changes to the atmospheric composition of $O_3$ precursors, particularly with respect to CO, benzene, ethyne, the alkenes, and PANs. Additionally the smoke may added many $O_3$ precursors that we were not set up to measure (e.g. many OVOCs). Due to the nonlinear nature of $O_3$ chemistry, the different mix of precursors could have caused enhanced local $O_3$ production, depressed local $O_3$ production, or had no effect on local $O_3$ production. . Taken together, the observations do not suggest a single mechanism that describes smoke influence on $O_3$ in Front Range airmasses during these case studies. Instead, the observations point to the presence of smoke resulting in a complex array of processes that will require more detailed observations and chemical transport modeling to clearly identify and quantify."*

Figure 3. Out of the 40+ VOCs measured, why were these compounds chosen when most of them have other large sources in the area? Instead of the binary color scheme, can a colorscale be applied to show the percent change for each species?

*These were species that showed significant changes between the August smoke-impacted period and the smoke-free period, which were the two periods during which valid VOC data were collected. The authors feel that a percent change colorscale would make this figure too complex to digest. We have referred readers to the summary of the full VOC dataset in Table S1 for specifics.*

Figure 5: Do the color bands represent one standard deviation of each average diurnal cycle?

*Yes, the shading represents one standard deviation. The figure caption has been amended to say this.*

Technical corrections: Line 94 (and elsewhere): "1 $\mu$m PTFE filter membrane" Do you mean the pore size, not the filter size, was 1 um?

*The reviewer is correct, the filter pore size is 1 $\mu$m. The text has been corrected. Other sentences that included a reference to filter size were likewise corrected.*

*The inlet was located 6 m above ground level (a.g.l.), and a PTFE filter membrane with 1 $\mu$m pore size (Savillex) at the inlet was changed weekly.*

Figure 1: Please include more tick marks on the date axis so that specific dates can be located on the traces.

*Tick marks have been added identifying every 7 days in Figure 1, starting at the first of each month. More tick marks become crowded and distracting to the main point of the figure, which is to identify the smoke-impacted periods. The dates for these periods are labeled at the top of the figure and are specified in the text. The updated figure is below.*

[Figure]

Figure 6: Can Fig. S4 be merged with this one so all of the data is included in a single plot?

*This can be done. The updated Figure 6 is below.*

[Figure]

Figures S8 and S9: Arrange the panels in the same order.

*Arrangement updated so that Figure S9 matches Figure S8. New Figure S9 is shown here.*

[Figure]

Figures S9 and S10: Note more clearly in the caption which data are shown (e.g., afternoon only?). Also include labels for the data in the legend, not just the number of points.

*We have updated the caption and made the requested changes to the original Figures S9 and S10.*

References:

Akagi, S. K., Yokelson, R. J., Wiedinmyer, C., Alvarado, M. J., Reid, J. S., Karl, T., Crounse, J. D., and Wennberg, P. O.: Emission factors for open and domestic biomass burning for use in atmospheric models, Atmos. Chem. Phys., 11, 4039-4072, DOI 10.5194/acp-11-4039-2011, 2011.

Atkinson, R., and Arey, J.: Atmospheric Degradation of Volatile Organic Compounds, Chemical Reviews, 103, 4605-4638, 2003.

de Gouw, J. A., Cooper, O. R., Warneke, C., Hudson, P. K., Fehsenfeld, F. C., Holloway, J. S., Hubler, G., Nicks, D. K., Nowak, J. B., Parrish, D. D., Ryerson, T. B., Atlas, E. L., Donnelly, S. G., Schauffler, S. M., Stroud, V., Johnson, K., Carmichael, G. R., and Streets, D. G.: Chemical composition of air masses

transported from Asia to the U. S. West Coast during ITCT 2K2: Fossil fuel combustion versus biomass-burning signatures, J. Geophys. Res. Atmos., 109, Artn D23s20 10.1029/2003jd004202, 2004.

Heilman, W. E., Liu, Y. Q., Urbanski, S., Kovalev, V., and Mickler, R.: Wildland fire emissions, carbon, and climate: Plume transport, and chemistry processes, Forest Ecol Manag, 317, 70-79, , 2014.

Jaffe, D., Bertschi, I., Jaegle, L., Novelli, P., Reid, J. S., Tanimoto, H., Vingarzan, R., and Westphal, D. L.: Long-range transport of Siberian biomass burning emissions and impact on surface ozone in western North America, Geophys. Res. Lett., 31, Artn L16106 Doi 10.1029/2004gl020093, 2004.

Mauzerall, D. L., Logan, J. A., Jacob, D. J., Anderson, B. E., Blake, D. R., Bradshaw, J. D., Heikes, B., Sachse, G. W., Singh, H., and Talbot, B.: Photochemistry in biomass burning plumes and implications for tropospheric ozone over the tropical South Atlantic, J. Geophys. Res. Atmos., 103, 8401-8423, 10.1029/97jd02612, 1998.

Wotawa, G., and Trainer, M.: The influence of Canadian forest fires on pollutant concentrations in the United States, Science, 288, 324-328, 10.1126/science.288.5464.324, 2000.

---

## Author Comment (AC1)

MS No.: acp-2017-171

*We thank the reviewer for their time and thorough review of our manuscript. We have tried to address every comment and feel that doing so has resulted in a much improved manuscript. Our responses to the reviewer comments are below in italics, with text excerpts in quotes.*

General Comments: Overall, this is a good example of a case study, with relevance specifically to Colorado air quality and which, more generally, speaks to air quality concerns relevant to the western United States, where wildfires are a substantive air quality issue. In Colorado, there is an active community of regulators at the State and Federal level who have been debating the very issues discussed in this paper for well over a decade. There is an extensive network of monitoring and also substantive photochemical modeling address this issue from a policy perspective. This case study has merit, but the danger here is that a paper will over generalize a case study and overstate its own importance. The authors should be careful in this regard. From a regulatory perspective, actual exceedances of ambient standards for ozone (70 ppbv for 8 hours) are relevant and anything below this is generally not relevant. Even so, a non-attainment designation is based on more than a single exceedance at a single location. The authors should demonstrate that knowledge and perspective in the body of this work. It is well established that wildfire smoke can enhance ozone formation, especially air masses that have been aged for several days. Despite an extensive monitoring network and a concentrated field study, measurements are sparse, as is evidenced here by the use of data from several monitors remote from the BAO tower. This paper would be strengthened immensely by the use of satellite imagery showing the wildfire smoke haze during the periods of interest and also by the use of additional State and Federal agency data to establish that wildfire smoke was the cause of the ozone enhancements observed. In my mind, this is not unequivocally established by the observations presented here. For example, could CO enhancements be caused by Asian airmass transport? I encourage the authors of this work make substantial improvements to this work before I recommend publication of this work in Atmospheric Chemistry and Physics. Also, in this work, the authors contrast data from two "smoke" events with that from nonimpacted periods during the same period. However, much of the data from the first fire plume is disregarded. It's excluded from much of the text and the figures. In my opinion, the first fire plume should be included in all analyses, even if the results are diminished. If this is really a fire plume, it should not be dismissed and excluded. Fire plumes are variable, and that is an important point. Sometimes they make a case easier, sometimes more difficult, but this is a reality in a complex world.

*We thank the reviewer for their perspective here. The strength of this paper is that it shows two examples of how a subset of ozone precursors changes in the presence of aged fire smoke. We are not able to probe changes in composition as extensively for the smoke-impacted period in July because that technically occurred before the start of our field campaign. We were fortunate that many of the "easy" measurements (i.e. ozone, CO) were running already at that time, but the more labor-intensive instruments (i.e. the gas chromatographs used for the VOC measurements) were not running. The dataset is interesting because of the high quality of the observations, but it is also interesting because the fires responsible for the smoke in August 2015 were extreme. The 2015 Washington wildfires season was the largest in history. There are a number of case studies, with high chemical specificity, of aged wildfire smoke. However, there are very few measurements of this duration (i.e. aircraft will only sample a plume over the course of a few hours) or within a polluted boundary layer. This paper does very carefully demonstrate that ozone during both the July and August smoke-impacted periods was higher than expected based on ambient temperatures (i.e. for a given temperature average hourly ozone is greater during the smoke-impacted periods than the smoke-free period). However, more importantly, it shows which ozone precursors also change in the presence of smoke. We do not understand the mechanisms driving all these changes. However, there are other papers demonstrating that state of the science air quality models cannot always reproduce observations of elevated ozone when smoke impacts urban areas (e.g. Singh et al., 2010). Our manuscript is an important contribution to our understanding of how aged smoke impacts air pollution mixtures, and our target audience is comprised of atmospheric chemists. In response to the reviewer's comment that this paper is aimed at explaining ozone exceedances, we have revised the discussion substantially. Specifically, we now use the 95$^{th}$ percentile, rather than an MDA8 value, to subset elevated ozone. We agree with the reviewer*

*that satellite data is essential for validating our attribution of smoke periods – and this is exactly why we used the HMS smoke product, which is in fact based on satellite data. As this use of satellite data may have been unclear in our initial manuscript, we have substantially increased our explanation of that product. Finally we respectfully disagree that the CO enhancements observed in August 2015 over Colorado could have been due to transpacific transport. We present multiple lines of evidence that these enhancements were associated with the wildfires in Washington, as does Creamean et al. [2016 ACP].*

Specific Comments
1) The Title and Abstract should more strongly indicate that this is a single case study showing influences from remote wildfire smoke on one location downwind

*The authors agree that the title and abstract can be edited to be more specific. The title was revised to: "Changes in ozone and precursors during two aged wildfire smoke events in the Colorado Front Range in summer 2015".*

2) The Abstract should mention if nearby official monitors showed ozone exceedances to put this case study into context (see additional comments to this effect below).

*As discussed above, our aim is not to identify exceptional events. Rather our goal is to carefully document significant changes in ozone and its precursors associated with the presence of smoke using high-quality observations. We believe that the most easily accessible summary for interested readers on ozone exceedances is available through the Regional Air Quality Council (https://raqc.egnyte.com/dl/PwqCfyKZHM/2015%20Ozone%20Season%2010-21.pdf_), As we have tried to re-focus the introduction on the significance of these wildfire events, we have added the following information in the discussion of the ozone timeseries (Figure 7) rather than in the introduction.*

*"Several Front Range $O_3$ monitors recorded elevated ozone during the August smoke-impacted period. Specifically, the maximum daily 8-hour average ozone mixing ratio at Aurora East exceeded 75 ppbv on 21 August. This was the first highest maximum for this station for summer 2015. The second highest maximum for summer 2015 coincided with the August smoke-impacted period at Fort Collins West, Greely, La Casa, Welby and Aurora East. The third highest maximum for summer 2015 coincided with the August smoke-impacted period at Aurora East, South Boulder Creek, Rocky Mountain National Park, and Fort Collins – CSU."*

3) Introduction. I recommend that the authors extend the background discussion to include policy relevant discussions and demonstrate knowledge of the extensive regulatory work that is ongoing on this topic in the west related to ozone exceedances. For example, only one exceedance of the NAAQS for ozone occurred for all of July and August at the measurement site. How does this compare to the exceedences for the entire State for 2015? Was the entire State in non-attainment this year? Was the event mentioned here a contributing factor? Or, did it fall much farther down the list other exceedences of the standard in the Front Range in 2015. These questions are very relevant to policy and should be discussed in some detail to place the study in a larger context.

*We reiterate that the BAO ozone monitor is not an EPA Air Quality Monitor, and thus it is not used to determine ozone exceedances. We hesitate to add a comprehensive discussion of ozone exceedances for Colorado for 2015 as this will serve to focus the paper on policy, rather than atmospheric chemistry. Our aim is to show detailed chemical composition changes associated with the presence of aged smoke in the Front Range. However, we have added very specific information on which Front Range ozone monitors recorded elevated ozone during the smoke-impacted periods (see response to comment above). We are currently working on a second manuscript that provides detailed analysis of the elevated ozone observed at BAO that was not associated with the presence of smoke.*

*4) Introduction/Smoke events: How does the climatology of the study period compare to Colorado's as a whole? Was it a cool year? A hot year? A wet year? Was there evidence of pollution transport besides fire smoke from other areas of the US and Internationally?*

*The key point is that the Washington 2015 wildfires were extreme. They were the largest in that state's history. We have added this information to the introduction. As a specific response to this suggestion, we have also added the following sentences to the manuscript.*

*"Front Range surface temperatures were not anomalously high in July and August 2015 based on a comparison of reanalysis data for this period to a 1981 – 2010 climatology. Surface precipitation, surface relative humidity, and soil moisture in the Front Range were all lower than this referent period. The extreme fires in Washington and Idaho were associated with warmer and dryer than average summer temperatures in the Pacific Northwest (Kalnay et al., 1996)."*

*Kalnay, E. and Coauthors, 1996: The NCEP/NCAR Reanalysis 40-year Project. Bull. Amer. Meteor. Soc., 77, 437-471.*

*We have not identified other clear transport events in our dataset for 2015 at this time.*

5) It is not evident why values from the study site were compared with very select other monitoring sites (in this case CAMP, ROMO and Walden) and not others. This gives the impression that supporting evidence has been picked rather than evaluated broadly. Why, why for example, is the PM from the CAMP site (20+ miles from the BAO tower) used, while the CO measurements from CAMP have not? Why are other Front Range ozone measurements not evaluated? Why were Walden and Rocky Mountain ozone sites used, while other data (e.g. CO data from the Storm Peak Lab) were not? This needs to be address directly in the text.

*Thank you for pointing out that all these choices seemed arbitrary. We have edited the text to make our criteria for selecting other sites besides BAO clearer.*

*BAO, ROMO, and Walden are on a gradient of more to less anthropogenic influence. We included measurements from ROMO and Walden in Figure 8 to illustrate that the ozone enhancements are observed in locations outside the Front Range. To our knowledge Storm Peak does not have regular CO measurements during this time period.*

*We did examine CO measurements from CAMP, and they do show an enhancement in ozone. Median CO during the smoke-impacted periods is 500 ppbv, as compared to 300 ppbv during the smoke-free periods. However, the CO measurement at CAMP is less precise than that at BAO, and thus this measurement is less ideal for identifying the exact start and end of the smoke-impacted periods.*

6) Was there satellite imagery from this period that showed the wildfire smoke haze? This data is often widely available, and so should be included if possible.

*The HMS smoke product uses data from multiple NOAA and NASA satellites to identify smoke-plumes in the atmospheric column The smoke is detected using visible imagery assisted by infrared imagery, which allows clouds and smoke to be distinguished. A full description of the HMS smoke product is available in Brey et al. [2017], currently under review in ACPD. We have added this information to the text, see below.*

*"The NOAA Hazard Mapping System smoke polygons (grey shading) show that the smoke events observed at BAO were large regional events. The HMS smoke product is produced using multiple NASA and NOAA satellite products (Rolph et al., 2009). Smoke in the atmospheric column is detected using both visible and infrared imagery and is fully described in Brey et al. (2017). The extent of smoke plumes within the HMS dataset represents a conservative estimate, and no information is provided on the vertical extent or vertical placement of the plumes."*

*Brey, S. J., Ruminski, M., Atwood, S. A., and Fischer, E. V.: Connecting smoke plumes to sources using Hazard Mapping System (HMS) smoke and fire location data over North America, Atmos. Chem. Phys. Discuss., https://doi.org/10.5194/acp-2017-245, in review, 2017.*

*The presence of smoke is also supported by lidar measurements from CALIPSO. Creamean et al. (2016) used CALIPSO data to investigate aerosol composition during the August smoke period. Below we have provided figure showing a CALIPSO overpass through the Front Range, close to BAO, and this data also shows clear contributions of wildfire smoke to the detected aerosol. For example black and red colors both represent possible smoke contribution to the aerosol detected by CALIPSO throughout the column. The plot also shows that smoke aerosol extends from the ground (the base of all the colors roughly follows the contours of the surface elevation) to the mid troposphere. It is clear that CALIPSO is sampling the widespread regional smoke plume that is also seen in the HMS smoke product during this same time period.*

[Figure]

**Aerosol Subtype  UTC: 2015-08-24 09:06:50  Version: 4.1  Nighttime**

0 = N/A    1 = marine    2 = dust    3 = polluted continental/smoke    4 = clean continental    5 = polluted dust    6 = elevated smoke

7) Line 102. The GC-method needs to be summarized in more detail. What is the integration period? What is the frequency of measurement? The overall method can be referenced from another paper, but those parameters are important and should be included here.

*We have provided answers to the reviewers questions as edits to the text and continue to point to the full description of the GC instrument in Abeleira et al. (2017). The revised text is below.*

*"A custom 4-channel cryogen-free gas chromatography (GC) system (Sive et al., 2005) was used to measure selected non-methane hydrocarbons (NMHCs), $C_1 – C_2$ halocarbons, alkyl nitrates (ANs), and oxygenated volatile organic compounds (OVOCs) at sub-hourly time resolution; approximately one sample every 45 minutes. The inlet was located at 6 m a.g.l. with a 1 μm pore size teflon filter. Ambient air for each sample was collected and preconcentrated over 5 minutes, with a one liter total sample volume. A calibrated whole air mixture was sampled in the field after every ten ambient samples to monitor sensitivity changes and measurement precision.  A full description of this instrument and the associated uncertainties for each detected species is provided in (Abeleira et al., 2017)."*

8) Line 147 and Figure 1. CO and PM2.5 data from other surrounding monitors should also be included. Ozone data from other Front Range Non-attainment area monitors should be summarized and discussed.

*Below we show a timeseries of daily average PM measurements for summer 2015 from 10 PM monitors in the Front Range: CAMP, BOU, CASA, CHAT, COMM, FTCF, GREH, I25, LNGM, NJH. All monitors show similar and consistent excursions during the same smoke-impacted time periods defined at BAO (shown in red shading).*

[Figure]

*In response to an earlier comment, we have added the following sentences on nearby ozone monitors in Section 4.3.*

*"Several Front Range $O_3$ monitors recorded elevated ozone during the August smoke-impacted period. Specifically, the maximum daily 8-hour average ozone mixing ratio at Aurora East exceeded 75 ppbv on 21 August. This was the first highest maximum for this station for summer 2015. The second highest maximum for summer 2015 coincided with the August smoke-impacted period at Fort Collins West, Greely, La Casa, Welby and Aurora East. The third highest maximum for summer 2015 coincided with the August smoke-impacted period at Aurora East, South Boulder Creek, Rocky Mountain National Park, and Fort Collins – CSU."*

*The CO data at the other monitors are substantially less precise than that we collected at BAO, but all showed enhancements during the July and August smoke-impacted periods of between 60 and 150%, the same range of percentage increases as observed for CO at BAO.*

9) Figure 1. Add CO from CAMP at minimum. There are other CO monitors nearby, do they agree? Add PM2.5 from locations closer than CAMP if possible. Address the latter points in text if they cannot be added to the figure.

*See above.*

10) Lines 147-157. CAMP also has O3. The data from that station's O3 needs to be included/shown here, and any place PM2.5 is used from that site, especially given the 35 km distance between sites. Also, there needs to be a space between 35 and km on line 150. The paper states, "PM2.5 was similarly elevated during the smoke-impacted periods at CDPHE monitoring sites across the Colorado Front Range (not shown)." Why is it not shown? It should be. Lastly, the spikes in figure 1 data are of equal magnitude as the spikes within the defined periods, why are these smoke free?

*It is unclear why the reviewer is suggesting that we emphasize the CAMP site. This is just one of 14 ozone monitors that were operational in summer 2015 in the region. We have added a very specific description of which sites show elevated ozone during the fire impacted period based on Regional Air Quality Council analysis available here: https://raqc.egnyte.com/dl/PwqCfyKZHM/2015%20Ozone%20Season%2010-21.pdf_*

*Yes, there are "spikes" in CO throughout the campaign; however, these are not accompanied by large increases in aerosol concentration and tend to be of very short duration (on the order of minutes). The enhancements in CO and PM2.5 during the smoke events are well correlated and last for hours to days.*

11) Figure 2. Recommend that satellite imagery of smoke added as additional figure to make the case that the plume was smoke and widespread.

*As discussed in response to an earlier comment, the HMS smoke product (shown in Figure 2) uses data from multiple NOAA and NASA satellites to identify smoke-plumes in the atmospheric column The smoke is detected using visible imagery assisted by infrared imagery, which allows clouds and smoke to be distinguished. A full description of the HMS smoke product is available in Brey et al. (2017), currently under review in ACPD. We have added this information to the text, see below.*

*"The NOAA Hazard Mapping System smoke polygons (grey shading) show that the smoke events observed at BAO were large regional events. The HMS smoke product is produced using multiple NASA and NOAA satellite products (Rolph et al., 2009). Smoke in the atmospheric column is detected using both visible and infrared imagery and is fully described in Brey et al. (2017). The extent of smoke plumes within the HMS dataset represents a conservative estimate, and no information is provided on the vertical extent or vertical placement of the plumes."*

*Brey, S. J., Ruminski, M., Atwood, S. A., and Fischer, E. V.: Connecting smoke plumes to sources using Hazard Mapping System (HMS) smoke and fire location data over North America, Atmos. Chem. Phys. Discuss., https://doi.org/10.5194/acp-2017-245, in review, 2017.*

*We also note the CALIPSO data described in the answer to an earlier comment.*

12) Section 4.1. Measured data, especially for VOCs, should be tabulated and summarized. Please insert a relevant table of species measured with relevant max, min, median values and standard deviations.

*We have added a table in the SI that provides this information.*

13) Figure 3. Labels not clear. Add text labels. Why is first fire period excluded?

*We have added further labels to indicate that red signifies significant positive changes in the VOCs during the smoke-impacted period, and that blue signifies significant decreases in the VOCs during the smoke-impacted period. There are no VOC measurements during the July fire period. As discussed in response to an earlier comment, the first fire period occurred before our planned field intensive. We had simple instrumentation running (i.e. ozone, CO measurements), but the labor intensive GCs used for the VOC measurements were not running at this time.*

14) Figure 3 is hard to understand without a table or plot of VOC concentrations.

*We have added a table in the SI that provides this information.*

15) Lines 178-181. The fact no biomass burning specific VOC speciation was done at all seems a bit off. This is surprising given the title, and conclusions, and likely impacts of this paper.

*As discussed in the paper, the motivation for this field campaign was not associated with biomass burning. It is actually very difficult to plan such an experiment. The campaign was planned long before summer 2015, and was not initiated in response to the smoke. The GC was not optimized to be sensitive to biomass burning specific tracers such as HCN or acetonitrile. Were we to know we would get to sample this type of natural experiment again we would certainly make an effort to include measurements of such species.*

16) Line 188. Section 4.1, Figure 2: Unclear how the statement ". . . suggests that the age of smoke impacting the Front Range during the August smoke-period was 2-3 days." That is not apparent in the figure.

*Thank you for noting that this was unclear – it looks like we accidentally lost some information from the caption for Figure 2. We have added the following text to clarify this point.*

*"The HYSPLIT backward trajectories shown in Figure 2 are 5 day backward trajectories."*

*We have also added this to the caption for Figure 2, and we have added 24-hour markers to the trajectories plotted in this Figure.*

17) Figure S1-S2. These plots are jumbled. Add legends. If your point is that the boundary layers at 0Z are more variable than the ones at 12Z, you should make that point in the text. The data contradict the conjecture you make around line 218. It's not clear why the sondes are included here. The surface temperature data is presented in Figure 8, so why show the sondes? Perhaps these figures should be revised to be simpler and more concise, or removed. If you must show the soundings then perhaps have two panels, one for smoke free and one for smoke and then a solid gray-area representing all the data, and a line for the average, or even a vertical box/whisker plot.

*We agree with the reviewer that these figures are not necessary, and we had just included them for completeness. We have removed them from the supplemental material.*

18) Line 216. "Not shown" in reference to the diurnal cycles. Diurnal cycles should be shown. 19) Line 218. A lower PBL height during the day is exactly the opposite of what is observed and this directly contradicts data from figures S1 and S2. This speculation should be removed.

*Showing all the diurnal cycles associated with the VOC species would make an unmanageable number of figures, even for the SI. The changes to the alkanes were largely insignificant. We agree that this is a confusing detail, and with the removal of S1 and S2 as suggested above, we have also removed this text. The key point is the very high abundance of alkanes in our region, we agree that the discussion of the diurnal cycles is distracting and we have removed this section of text. In response to a later comment about diurnal cycles, we have shown them for the alkenes in the SI.*

20) Line 231-2. This statement does not make sense. Abundances decrease over what time period? Please clarify the wording.

*Thank you for pointing this out. This sentence has been clarified and pasted below.*

*"Surprisingly, we observed significant decreases in the abundance of isoprene, propene and ethene during the August smoke-impacted period compared to the smoke-free period: -64% (-143 pptv), -77% (-39 pptv), and -81% (-206 pptv) respectively (for summary statistics see Table 1)."*

21) Line 232. Diurnals not shown. Conclusions in this section could use the support of the diurnal cycles and as is it's hard to follow without them.

*We agree that this is a better use of SI figures than the soundings, so we have added plots of the diurnal cycles for the impacted alkenes (isoprene, propene and ethene) to the SI in Figure S1.*

22) Paragraph including lines 230-246. Trends are not explained and speculation here is spurious. A table presenting the measured values could easily replace this table. Also, why would isoprene behave differently than other alkenes? Are the changes in alkenes even significant given that they are near their detection limits? A table would suffice here rather than trying to explain trends in ways that mean little. It is unclear what the conclusion of this paragraph is. It's also unclear what the take home point is or and how the evidence supports the conclusion.

*We agree that we do not have an irrefutable mechanism to explain this observation, but the changes that we observed are significant. The mixing ratios were substantially suppressed, such that they were near their detection limits. To our knowledge, this is the first time this has been observed. This is very interesting because the aged smoke clearly changed either local emissions or oxidation rates in some way. To improve this paragraph, we have added a table as suggested by the reviewer. We have edited this section and have pasted the revised version below.*

*"The atmospheric lifetimes of the four alkenes we quantified (isoprene, propene, ethene, and cis-2-butene) range from tens of minutes to hours. Surprisingly, we observed significant decreases in the abundance of isoprene, propene and ethene during the August smoke-impacted period compared to the smoke-free period: -64% (-143 pptv), -77% (-39 pptv), and -81% (-206 pptv) respectively (for summary statistics see Table 1). The shape of the diurnal cycles did not change (Figure S1), though propene and ethene were near their respective limits of detection for the majority of each day during the smoke-impacted period. Given the short lifetimes of these species, this indicates that the presence of the smoke changed either local anthropogenic or biogenic emissions of these species, or their respective rates of oxidation by OH or $O_3$. We present several potential mechanisms here, but we do not have sufficient information to determine if one of these is solely responsible for the pattern we observed.*

*Our first hypothesis is that fewer anthropogenic emissions of these alkenes drove the observed decreases in alkene abundances. However, there is no evidence that anthropogenic emissions were different during the August smoke-impacted period. Specifically, the August smoke-impacted period encompassed both weekdays and weekends and did not contain any state or federal holidays. Therefore we move to our second hypothesis, that changes in the biogenic emissions of alkenes accounted for the decreased alkene mixing ratios. Isoprene is widely known to be emitted by broad leaf vegetation, and emission rates are positively correlated with light and temperature (Guenther et al., 2006). Recent measurements quantified ethene and propene emissions from a ponderosa pine forest near Colorado Springs, CO, with an inter-daily light and temperature dependence similar to isoprene (Rhew et al., 2017). Interestingly, emissions and mixing ratios of ethene and propene were not closely correlated with isoprene within the diurnal cycle, indicating they have different vegetative/soil sources than isoprene at that site. Ponderosa pine stands are present in the foothills on the western edge of the plains in the Front Range, and several species of broad leaf trees are present along waterways, in urban areas, and in the foothills of this region. Thus, biogenic sources of ethene, propene, and isoprene in the region around BAO are reasonable. Given the August smoke-impacted period was on average colder than the smoke-free period, and potentially saw a reduction in photosynthetic active radiation (PAR) at the surface due to the increased number of aerosols, it is possible that biogenic emissions of isoprene, ethane, and propene were suppressed. However, biogenic fluxes of these compounds are unavailable for the region around BAO during summer 2015, and extrapolating emissions from one ponderosa pine stand to the rest of the Front Range may be overly ambitious. Further, we note that a PMF analysis of the VOC data from this site did produce a 'biogenic factor' dominated by isoprene, but with negligible contribution of any other hydrocarbon, suggesting that the biogenic component of these $C_2$-$C_3$ alkenes was small (Abeleira et al...). Thus, while the hypothesis that smoke suppressed biogenic emissions remains feasible, we will consider other potential causes for the observed decrease in alkene abundances.*

*The alkenes we measured all have high reactivities with respect to OH (> 8 x 10$^{12}$ molec$^{-1}$ cm$^3$ s) and O$_3$ (> 0.1 x 10$^{17}$ molec$^{-1}$ cm$^3$ s) (Atkinson and Arey, 2003). Enhancements in OH abundances have been inferred in wildfire smoke plumes by several studies (e.g. Akagi et al. (2012); Hobbs et al. (2003); Liu et al. (2016); Yokelson et al. (2009)). If the August smoke-impacted period was characterized by higher than normal OH mixing ratios, then a third hypothesis is that the observed decreases in alkene abundances could be due to a higher oxidation rate by OH due to higher OH concentrations. However, other measured VOCs such as o-xylene or methylcyclohexane have similar OH reactivities to ethene (Atkinson and Arey, 2003), and we do not see associated decreases in abundances of these other VOCs. Thus, the hypothesis of increased oxidation by OH causing decreased alkene abundances in the August smoke period is not supported by the full suite of measurements at BAO.*

*Lastly, we move on to our final hypothesis. Alkenes have much higher rates of reaction with O$_3$ than the other VOCs we quantified. As we will demonstrate in Section 4.3, the August smoke-impacted period was characterized by higher O$_3$ abundances than would otherwise be expected. Therefore, the fourth hypothesis regarding decreased alkene abundances is that enhanced alkene oxidation by O$_3$ decreased the observed mixing ratios. Two factors complicate this hypothesis though. First, we do not observe a negative relationship between O$_3$ and alkene abundance during the smoke-free time periods (i.e. increased O$_3$ is not correlated with decreased alkenes when no smoke is present). Second, despite having a higher reaction rate with O$_3$ compared to propene and ethene, cis-2-butene does not decrease during the August smoke-impacted period.*

*After careful consideration, there is no strong evidence supporting any of these four hypotheses over the others (suppressed anthropogenic emissions, suppressed biogenic emissions, increased OH, increased O$_3$). It is possible that more than one of these processes could have contributed to the observation of decreased alkene abundances during the 2 week-long August smoke-influenced period. Future field campaigns and modeling work are necessary to understand how common suppressed alkene abundances may be in smoke-impacted airmasses, and what processes might control this phenomenon."*

23) Figure 4. Include first fire period. This figure does not appear to be referred to in the text? It is unclear what 95th percentiles mean. In the legend says quantile and not percentile. Clarify. If it is not referred to in the text, it should be eliminated.

*The legend has been fixed and the caption amended to clarify the meaning of 95$^{th}$ percentiles. This Figure is already referenced to in line 273 of the original manuscript.*

24) Figure 5. Indicate what shaded regions are. Are they percentiles? Of which measurements? Note that almost never does red line leave the grey shaded area, except for PAN and NOx. Discuss in text. Show solar noon on the plots for clarity.

*We have tested the significance of the differences using a 2-tailed Student's t-test at the 95% confidence level, which describes the likelihood that two sets of data come from the same population. Shaded areas represent one standard deviation (67%) of a single population, assuming a normal distribution; overlap between standard deviations is not typically a metric for two datasets coming from the same population. The text has been edited to clarify this point. Solar noon varies throughout the summer, but the changes are quite small over our time period, and we have compared our data by hour. We have added this information to the caption:*

*"Solar noon on 1 July 2015 was at 1:03 PM, solar noon on 7 September was 2015 was at 12:57 PM."*

25) Line 308. Please include more detail about the analysis you did related to traffic impacts.

*In responding to this request we re-evaluated the analysis we had done previously, and took another detailed look at the time series. Previously we had searched for any consistent patterns in wind direction or speed during the large NO$_2$ peaks observed in the August smoke-impacted period, as well as looked at the correlation of NO$_2$ with NO. Our assumption was that since I-25 is within 2 miles of the BAO site that large NO$_2$ peaks coming from I-25 traffic would be freshly emitted NO$_x$ and therefore closely correlated with NO. We did not find any consistent wind direction or correlation with NO, thereby we concluded that these*

*peaks were not related to traffic emissions. In revisiting this analysis we considered each large increase in $NO_2$ individually. We have added the following text to the manuscript to describe this additional analysis. We also pose an additional hypothesis for the changes that we observed that was suggested to us when this work was presented earlier this last month.*

*"Out of 7 morning peaks in $NO_2$ during the August smoke-impacted period, 3 had concurrent toluene and ethyne peaks. One of these days occurred on a weekend, and the others occurred on weekdays. Toluene and ethyne are common tracers of traffic/industrial emissions. However, 4 of the days did not have corresponding ethyne and toluene peaks. Thus, traffic may have impacted some of the $NO_2$ enhancements we observed, but there is also likely another contributing mechanism. There are a few potential hypotheses for a non-traffic related $NO_2$ enhancement during the August smoke period. One hypothesis is that the photolysis frequency ($J_{NO2}$) was most impacted (i.e. reduced) by the smoke near sunrise and sunset."*

26) O3 does indeed have a positive correlation with surface temperature as referenced often in this paper. However for the Front Range region, this should be tempered by the fact that the almost parallel rise in temperature and ozone starts dropping off after the air temperature hits about 86-90F (30C-32C). Some evidence of this can be seen in Figure 6. The reason for this is that once surface temperatures begin to exceed this threshold, a westerly wind component usually becomes dominant. These westerlies will often be gustier and not allow the cyclical terrain-driven circulations that normally enhance ozone concentrations across the Front Range. As referred to in this paper, the Reddy-Pfister study of 2016 expands on this and concludes that 500 mb heights and 700 mb winds hold a stronger correlation to ozone concentrations than surface temperature for Front Range locations. Perhaps this is irrelevant since the air temperature during the "smoke" periods did not get very hot, but maybe an explanation of this phenomena should be included if surface temperature is being emphasized as being more important than the other variables mentioned above.

*Thanks for this note. We are happy to provide this context, and we already had supplemental figures showing the lack of correlation between 500 mb heights and 700 mb temperatures with MDA8 at BAO during 2015. This is not necessarily inconsistent with the Reddy and Pfister conclusion, as a key difference is that Reddy and Pfister's conclusions are based on the interannual variability of monthly average conditions. There are also other chemical factors that could be contributing to this pattern, including a shortened thermal lifetime of PAN. This section of text now reads.*

*"$O_3$ mixing ratios generally increase with temperature, and this relationship has been attributed to several specific processes including 1) warm and often stagnant anti-cyclonic atmospheric conditions that are conducive to $O_3$ formation, 2) warmer air temperatures that reduce the lifetime of PAN, releasing $NO_2$, and 3) lower relative humidity that reduces the speed of termination reactions to the $O_3$ production cycle (Jacob et al., 1993; Camalier et al., 2007). Specific to the Front Range, Abeleira et al. (2017) show that ozone in in this region has a temperature dependence, but it is smaller than other U.S. regions, consistent with the smaller local biogenic VOC emissions compared to many other locations in the eastern U.S. Finally, there is an additional meteorological factor in the Front Range that can impact the temperature dependence of ozone. Gusty westerly winds are often associated with high temperatures, and these winds serve to weaken or eliminate cyclical terrain-driven circulations that normally enhance ozone concentrations across the Front Range."*

*Later in the paragraph we note:*
*"The increase in $O_3$ mixing ratios during the August smoke-impacted period compared to the smoke-free period is present across the entire range of comparable temperatures."*

27) Figure 6. Include first smoke impacted period on this chart in another set of colored box and whiskers if they are different from the second smoke period. Also, in this figure, the gray bars are indistinguishable from the gray circles, they blend together. Perhaps use black bars instead of gray. The same is true for Fig 8, S4, and S9.

*We had originally included the July smoke-impacted period in the SI (originally S4). This figure also shows increased ozone at the low-end of the temperature distribution during the July smoke-impacted period, i.e.*

*it is consistent with Figure 6 from the second period, but there are lower ozone values and lower temperatures overall. We have combined Figures 6 and S4 into a new Figure 6. We have also outlined the boxes as suggested.*

28) Figure 7. The point highlighted in mid-August where temperature is low and O3 is high looks interesting, why is this not considered smoke influenced given the paper's hypothesis?

*All elevated ozone periods are interesting, however, this particular point does not occur during a period with elevated CO or PM2.5. Thus it is not smoke-impacted. We are working on another manuscript that will provide case studies of the elevated ozone events that were not associated with smoke. There is significant variability in the ozone temperature relationship in the Front Range, consistent with most other ground sites. We in no way intend to claim that all high $O_3$ events in the Front Range are linked to smoke. We have added this sentence to the conclusions to ensure that this is clear.*

*"This case study describes two distinct smoke events where the presence of smoke likely increased $O_3$ abundances above those expected by coincident temperatures. However, we do not intend to claim that all high $O_3$ episodes in the Front Range are caused by smoke, nor that smoke will always cause higher than expected $O_3$."*

29) Figure S3. Clarify if the data shown is for one or both smoke free periods. Show both, using different are lines, if they are different from each other. It is hard to see what's happening at lower values due to so many points. Or figure could be revamped showing quantiles with error bars and all data in gray behind.

*Data are only from the August smoke-impacted period. The PANs instrument was not operational during the July smoke-impacted period. We have removed this figure as it seemed to be confusing for the second reviewer, and does not show significant changes.*

30) Figure S4. Combine this figure with Figure 6.

*These figures have been combined. See our response to comment above.*

31) Line 323. Figure 5d doesn't appear to show a very significant difference in ozone between the black and red shaded areas. Perhaps, the figure needs to be edited to make the true difference clearer; otherwise, it seems overstated in the text.

*The purpose of the figure is to show the diurnal cycles of each species. To more easily visualize the differences in ozone abundances see Figure 6. The histogram in Figure 6 shows the difference in the distributions of all the data, and the boxplots show the difference as a function of temperature. As discussed above, we have tested the significance of the differences using a 2-sided Student's t-test. Significance is not indicated by non-overlapping standard deviations. Shaded areas are one standard deviation. The text has been edited to clarify this point.*

32) Line 330-333. O3 production with temperature levels off at high temperatures particularly in the Front Range due to the wind speed and direction associated with these high of temperatures. This should be addressed in the text.

*Thanks for this comment, it is similar to the one above, and we have addressed it through this modified text.*

*"$O_3$ mixing ratios generally increase with temperature, and this relationship has been attributed to several specific processes including 1) warm and often stagnant anti-cyclonic atmospheric conditions that are conducive to $O_3$ formation, 2) warmer air temperatures that reduce the lifetime of PAN, releasing $NO_2$, and 3) lower relative humidity that reduces the speed of termination reactions to the $O_3$ production cycle (Jacob et al., 1993; Camalier et al., 2007). Specific to the Front Range, Abeleira and Farmer (2017) show that ozone in in this region has a temperature dependence, but it is smaller than other U.S. regions, consistent with the smaller local biogenic VOC emissions compared to many other locations in the eastern U.S. Finally, there is an additional meteorological factor in the Front Range that can impact the temperature*

*dependence of ozone. Gusty westerly winds are often associated with high temperatures, and these winds serve to weaken or eliminate cyclical terrain-driven circulations that normally enhance $O_3$ mixing ratios across the Front Range."*

33) Lines 334-335. Things like black lines or red lines descriptions should be in the figure legend and caption, not text body.

*We thank the reviewer for catching this and have corrected the placement of the figure description.*

34) Line 361-363. The chosen altitude limit makes sense, but the Denver cyclone and in-basin wind patterns do contribute to ozone and recirculation. This should be emphasized more and discussed. The authors should include the wind field reanalysis data to show surface winds on their chosen day of interest in each smoke period.

*We agree that Denver cyclones and in-basin wind patterns do contribute to ozone production and re-circulation in the Front Range. We have added citations to two recent papers from the 2014 FRAPPE field campaign (Sullivan et al., 2016 and Vu et al., 2016), and more information on the two highest ozone days during the smoke-impacted period.*

*"Denver cyclones and in-basin wind patterns can also contribute to ozone production and re-circulation in the Front Range (see Sullivan et al. (2016), Vu et al., (2016) and references within). We examined surface wind observations (http://mesowest.utah.edu) on the highest ozone days during the smoke impacted period: 20 August and 25 August. There is no evidence of the establishment of Denver Cyclones on either of these days. Sullivan et al. (2016) point out that thermally driven recirculation can manifest as a secondary increase in ozone at surface sites. We did observe a secondary maxima at 17:00 MT on 25 August, but this feature was not present on 20 August."*

*Sullivan, J. T., et al. (2016), Quantifying the contribution of thermally driven recirculation to a high-ozone event along the Colorado Front Range using lidar, J. Geophys. Res. Atmos., 121, 10,377–10,390, doi:10.1002/2016JD025229.*

*Vu, K. T., Dingle, J. H., Bahreini, R., Reddy, P. J., Apel, E. C., Campos, T. L., DiGangi, J. P., Diskin, G. S., Fried, A., Herndon, S. C., Hills, A. J., Hornbrook, R. S., Huey, G., Kaser, L., Montzka, D. D., Nowak, J. B., Pusede, S. E., Richter, D., Roscioli, J. R., Sachse, G. W., Shertz, S., Stell, M., Tanner, D., Tyndall, G. S., Walega, J., Weibring, P., Weinheimer, A. J., Pfister, G., and Flocke, F.: Impacts of the Denver Cyclone on regional air quality and aerosol formation in the Colorado Front Range during FRAPPÉ 2014, Atmos. Chem. Phys., 16, 12039-12058, doi:10.5194/acp-16-12039-2016, 2016.*

35) Line 340. How did the weighting occur? Insert a reference or elaborate.

*The weighting is described in the text: "weighted by the total number of hourly measurements within each bin".*

36) Does the "synoptic scale transport" discussed at the end of page 11 and start of page 12 also account for the possibility of Asian pollution influence? The HYSPLIT back trajectories on page 20 both suggest that at least a portion of the air mass may have originated in Asia. It would be interesting to see just how much, if any, influence Asian pollution may have had when comparing the smoke and non-smoke air masses.

*We did not run backward trajectories of sufficient length to diagnose Asian transport. The transpacific transport of Asian pollution is more efficient in spring, though it can also occur in summer months. Diagnosing the contribution of Asian transport is beyond the scope of this paper, and would require the use (and careful evaluation) of a chemical transport model. We respectfully disagree with the reviewer that there is any evidence of Asian transport based on the data that we have.*

37) Line 352. It is unclear how just referencing the geopotential height paper (include citation at this location) leads to the conclusion that was "no evidence" of meteorological factors in ozone enhancement. This is a very broad generalization and needs supporting evidence and specific discussion if it is to be included here. Is the point you are making that the lack of meteorological factors that correlate with ozone implies that all the ozone was due to fire? If so, make this case strongly and state it clearly. Is absence of evidence meteorological driven ozone production even acceptable evidence? I'm not so sure it is. At best, it is supporting evidence.

*We agree that this wording might be confusing, and so have changed it to read:*

*"We tested the day-to-day variability in the relationship between $O_3$ and these meteorological variables during our study period using observations from the 0Z and 12Z atmospheric soundings conducted in Denver ([http://mesonet.agron.iastate.edu/archive/raob/](http://mesonet.agron.iastate.edu/archive/raob/)). The positive relationships between MDA8 $O_3$ and 700 mb temperature, 500 mb geopotential height, and surface winds are very weak, $R^2 = 0.04$, and $R^2 = 0.08$, and $R^2 = 0.0009$ respectively. Thus, we did not find any evidence to support the hypothesis that differences in meteorological conditions were solely responsible for the significant differences in composition or $O_3$ that we observed during the smoke-impacted period."*

38) Figure S5-7. The authors should explicitly discuss how the data in these figures supports their argument. This is a good supporting point, but there is need to flush out the discussion and figures better. Devoting 3 figures vague scatter plots to this is excessive. Could they be layered in 3 dimensions on a single plot? Alternatively, make one 3-panel figure or remove entirely and only quote the R2.

*Viewing these from the lens of a reviewer, we agree this is excessive. We have removed these figures from this version. We now only quote the $R^2$ as suggested by the reviewer. We have added the following sentence to the manuscript.*

*"The positive relationships between MDA8 O3 and 700 mb temperature, 500 mb geopotenial height, and surface winds are very weak, R2 = 0.04, and R2 = 0.08, and R2 = 0.0009 respectively."*

39) Line 368. Is this flow discussion where you should refer to Figure 4?

*No, this should not be a reference to Figure 4. This is correct that it should reference the original S9 and S10.*

40) Figure 7. This figure needs to be put into context. Did the Front Range exceed the NAAQS this year? Was this one of the four maximum values that put the region into non-attainment for the year? Or was it much further down the list? This is valuable context information that should be discussed in the text. One exceedence is generally irrelevant to the overall policy discussion, but if this is not the case, it is certainly worth discussing in more detail.

*We have added the following information to this section as suggested above.*

*"Several Front Range $O_3$ monitors recorded elevated ozone during the August smoke-impacted period. Specifically, the maximum daily 8-hour average ozone mixing ratio at Aurora East exceeded 75 ppbv on 21 August. This was the first highest maximum for this station for summer 2015. The second highest maximum for summer 2015 coincided with the August smoke-impacted period at Fort Collins West, Greely, La Casa, Welby and Aurora East. The third highest maximum for summer 2015 coincided with the August smoke-impacted period at Aurora East, South Boulder Creek, Rocky Mountain National Park, and Fort Collins – CSU."*

41) Paragraph lines 373-375. Why did you pick 65 ppbv MDA8, when this is not the standard? This seems arbitrary. Please use the current standard and put into the correct context of this year's ozone for the entire area as mentioned in a previous comment. Please also adjust your conclusions accordingly.

*In response to this concern and the one above, we have now re-framed everything in terms of 95th percentile ozone. This does not change any of our conclusions, the highest ozone days are still apparent regardless of the cutoff used.*

42) Figure 8. Same comment as for Figure 6. This needs to include the first fire period. Also, why were these sites chosen? Is it because they are remote? If so, why was an example of a nearby monitor not included? Was your point to show that the smoke was widespread? Pick more, not less sites. Was Colorado Springs impacted? It is not necessary to show all the sites, but just clarify your rational and pick sites to make your point and then say why you picked them.

*To be consistent with the revised version of Figure 6 (which now also includes the data plotted originally in S4), we have also added the July period to these plots. Ozone was not notably high in the July period. We picked the two sites because they are at different altitudes than BAO, and offer different information than additional surface sites within the polluted Front Range urban corridor. The RMNP site is often influenced by Front Range polluted air parcels in the afternoon, but not consistently throughout the day. The Walden site is largely free from Front Range influence – that is why it was chosen. The August smoke-impacted period at Walden also has higher ozone for a given temperature, and this is consistent with the hypothesis that at least a fraction of the ozone production within the August smoke plume occurred upwind of the polluted Front Range. This choice is explained in the following paragraphs which have been expanded to make this choice clearer.*

*"As mentioned in the Introduction, wildfire smoke can produce $O_3$ within the plume as it is transported, as well as contribute to $O_3$ photochemistry by mixing additional precursors into surface air masses. To assess the possibility of $O_3$ production with the plume, we analyzed hourly $O_3$ measurements from two National Park Service (NPS) Air Resources Division (http://ard-request.air-resource.com/data.aspx) measurement locations that are located outside the polluted Front Range urban corridor. The Rocky Mountain National Park long-term monitoring site (ROMO; 40.2778°N, 105.5453°W, 2743 meters A.S.L.) is located on the east side of the Continental Divide and co-located with the Interagency Monitoring of Protected Visual Environments (IMPROVE) and EPA Clean Air Status and Trends Network (CASTNet) monitoring sites. Front Range air masses frequently reach this site during summer afternoons (Benedict et al., 2013). The Arapahoe National Wildlife Refuge long-term monitoring site (WALD; 40.8822°N, 106.3061°W, 2417 meters A.S.L.) near Walden, Colorado, is a rural mountain valley site with very little influence from anthropogenic emissions. These two sites follow a rough urban to rural gradient; from primarily influenced by anthropogenic emissions (BAO), to sometimes influenced by anthropogenic emissions (ROMO), to very little influence from anthropogenic emissions (WALD). Figure 8 shows that the August smoke-impacted period produced increases in $O_3$ mixing ratios across all three sites. When comparing all data for a given temperature, there are average weighted enhancements of $10 \pm 2$ ppbv, $10 \pm 2$ ppbv, and $6 \pm 2$ ppbv $O_3$ at BAO, ROMO and WALD respectively. $O_3$ enhancements across all three sites, across an approximate urban to rural gradient, suggest that some amount of the $O_3$ enhancement observed at BAO during the August smoke-impacted period is the result of $O_3$ production within the plume during transit. $O_3$ during the July smoke-impacted period in Figure 8 shows a different pattern. As we saw in Figure 6, $O_3$ is enhanced above the level predicted by the ambient temperature at BAO. But no statistically significant enhancements are observed at ROMO and WALD for the July smoke-impacted period. One possibly reason for this nuance is that, based on the HMS smoke product shown in Figure 2, it is less obvious that smoke was present at ROMO and WALD during the July smoke-impacted period."*

43) Lines 414-416. What are you trying to say here? Can you refer to Figure 4? Are you trying to say the smoke was widespread? If so, say that and present evidence.

*Yes, we are referring to Figure 4 here. We have changed these sentences to read.*

*"We did not observe any consistent shifts in wind direction or changes in wind speed that can explain the observed changes in composition (e.g. Figure 4), and the changes in abundances that we observed for a given species were generally present across all directions and speeds. The smoke was ubiquitous across the Front Range as evidenced by enhanced $PM_{2.5}$ at CAMP and 9 other Front Range CDPHE monitoring sites."*

44) Line 424-6. You should state this much more strongly and earlier on. It is a major conclusion of the paper. You have direct evidence of this variability.

*We have changed this paragraph to read:*

*"It is important to note that the presence of smoke does not always result in very high $O_3$ abundances. Many other factors contribute to the overall level of surface $O_3$, and smoke can also be associated with relatively low $O_3$ at times, such as during the July smoke event described above. This case study describes two distinct smoke events where the presence of smoke likely increased $O_3$ abundances above those expected by coincident temperatures. However, we do not intend to claim that all high $O_3$ episodes in the Front Range are caused by smoke, nor that smoke will always cause higher than expected $O_3$. Each smoke event has unique characteristics and thus it is important to study and characterize more events such as these in the future."*

45) Figure S9-S10. Are these figure needed? Could they be combined with Figure 4 and used together as supporting evidence?

*These figures show different information than that contained in Figure 4. Figure 4 refers to local wind direction, whereas Figures S9 and S10 display the effect of long range transport and air mass history on the ozone temperature relationship. We have substantially reduced the number of supplemental figures in response to other comments – the original S8, S9, and S10 are the only remaining supplemental figures.*

46) Figure S11. Is this figure needed? Could you just state the values in the text? An entire figure for two data points with error bars is excessive.

*We have removed this figure, and added the values to the text.*

*"We do not find any significant differences in average calculated OPE between the smoke-impacted (8 ± 3 ppbv/ppbv) and smoke-free periods (7 ± 3 ppbv/ppbv )."*

47) References. Please include reference showing where the public data you used came from (CDPHE, Forest Service, NPS).

*We have verified a reference to the data source is in every place where the data are introduced, and added it if it was missing.*

Minor Issues/Typos

1) Line 62. The use of the pronoun "they" is vague. Please clarify the wording.
*Line 62 has been corrected to be more specific. The edited sentence is below.*

*"Brey and Fischer (2016) investigated the impacts of smoke on $O_3$ abundances across the U.S. via an analysis of routine in situ measurements and NOAA satellite products. Their analysis demonstrated that the presence of smoke is correlated with higher $O_3$ mixing ratios in many areas of the U.S., and that this correlation is not driven by temperature."*

2) Line 76. The term "This region" is vague and should be made more specific and the wording should be clarified.
*The sentence has been edited for clarity.*

*The Northern Colorado Front Range region violates the NAAQS for $O_3$, and has been the focus of several recent studies (e.g. McDuffie et al., 2016; Abeleira et al., 2017).*

3) Line 220. There should be a comma after "however"
*Corrected.*

4) Line 243. A comma is needed after "Thus"
*Corrected.*

5) Line 278. A comma is needed after "Thus"
*Corrected.*

6) Line 286. This is important and should be emphasized more if possible, rather than burying it deep in a paragraph. Perhaps making the PAN and alkyl nitrate discussions separate paragraphs would clarify enough.

*This appears to be very confusing for the reviewer and we apologize for this. We do not mean to imply that this ratio suggests that there is Asian influence. We meant to simply acknowledge that there is another example like this, where urban and biomass burning influenced ratios are compared, and this is the Roberts et al. [2004] paper. We have removed this sentence from the paper.*

7) Line 300. Phrasing is a bit confusing. Rather than saying "fewer days" which is a little vague here, rephrase saying that period 1 had a shorter duration than period 2, or the equivalent.

*This has been rephrased, and it now reads:*

*"…though the mixing ratios were within the range of smoke-free values and the duration of the July smoke-impacted period was much shorter than the August smoke-impacted period."*

8) Line 302. "more significant changes". . .than what? NO? Clarify the wording here.

*Yes, compared to NO. We have changed this sentence to read:*

*"Figure 5 shows that $NO_2$ abundances exhibited more significant changes than NO."*

9) Line 310. The phrase "is one hypothesis" is awkward. I suggest rephrasing this sentence.

*The part of the sentence containing this phrase was removed.*

10) Line 317. A comma is needed after "In this section"

*Corrected.*

11) Line 419. "Very high" is not specific enough. Include the value here.

*This conclusion has been made specific to the 95$^{th}$ percentile of 11am-4pm hourly ozone following the methodology of Cooper et al. (2012).*

*Cooper, O. R., R.-S. Gao, D. Tarasick, T. Leblanc, and C. Sweeney (2012), Long-term ozone trends at rural ozone monitoring sites across the United States, 1990–2010, J. Geophys. Res., 117, D22307, doi:10.1029/2012JD018261.*

12) P 6, line 161-166. Section 3: Things like red triangles, black lines, etc. should be in the figure caption, but not paper text. Only science/discussion should be in the paper body. Also, what are the blue circles? Not in legend or caption. There should be a space between 1000 and m on line 166.

*The authors thank the reviewer for catching these corrections. All suggested changes or clarifications have been made.*